# Configurational entropy and Adam-Gibbs relation for quantum liquids

Yang Zhou [1,2] ✉, Ali Eltareb[1,2], Gustavo E. Lopez[3,4] &
Nicolas Giovambattista [1,2,3] ✉

As a liquid approaches the glass state, its dynamics slows down rapidly, by a few orders of magnitude in a very small temperature range. In the case of light elements and small molecules containing hydrogen (e.g., water), such a process can be affected by nuclear quantum effects (due to quantum fluctuations/atoms delocalization). In this work, we apply the potential energy landscape (PEL) formalism and path-integral computer simulations to study the low-temperature behavior of a Lennard-Jones binary mixture (LJBM) that obeys quantum mechanics. We show that, as for the case of classical liquids, (i) a configurational entropy $S_{IS}$ can be defined, and (ii) the Adam-Gibbs equation, which relates the diffusion coefficient of a liquid and its $S_{IS}$, holds for the studied quantum LJBM. Overall, this study shows that one theoretical approach, the PEL formalism, can be used to describe low-temperature liquids close to their glass transition, independently of whether the system obeys classical or quantum mechanics.

Glasses, or amorphous solids, are out-of-equilibrium systems that can be formed by different routes, including cooling a liquid sufficiently fast so crystallization can be avoided[1–4]. The behavior of liquids close to the glass state, and during the associated liquid-to-glass transition have been the focus of numerous studies over the last few decades[5–8], and various theoretical approaches have been proposed to describe these systems[2,9–12]. In most theoretical/computational studies, the liquids/glasses are treated classically which is justified given that many common glass-formers, such as silica, are composed of heavy atoms and exhibit a high glass transition temperature (>1000 K). There are, however, liquids that vitrify at relatively low temperatures where nuclear quantum effects (due to the atoms delocalization) can play a relevant role. This is the case of small molecules that contain hydrogen, such as water[13], where isotope substitution effects are non-negligible even close to room temperature[14,15]. Obviously, nuclear quantum effects (NQE) play a relevant role in light-element liquids, such as $H_2$ and He (even at temperatures where classical statistics holds[16–18]). We also note that there are quantum systems, other than liquids, that exhibit a glass transition. These include electrons in

metals[19,20] and spin systems[21–23]. It remains unclear how quantum mechanics may change the behavior of low-temperature liquids close to their glass transition temperature, relative to the classical case. For example, an extension of mode coupling theory suggest that including quantum effects may increase the glass transition temperature of a liquid[24]. In other systems, quantum effects may erase the glass state altogether[25].

A well-established theoretical framework to study classical low-temperature liquids and glasses is the potential energy landscape (PEL) formalism[26,27]. The PEL formalism is based on statistical mechanics[28–30] and has been applied extensively to describe the behavior of atomistic[31–33] and molecular liquids[34,35]. The PEL formalism provides (i) a qualitative description of classical liquids/glasses, and (ii) under simple approximations, it also provides the Helmholtz free energy and equation of state of the system[35–40]. In previous studies[13,41,42], we extended the PEL formalism to the case of liquids that obey quantum mechanics. It was shown that (i') the same qualitative description of classical liquids and glasses, based on the PEL formalism, can be extended to quantum liquids. Moreover, path-integral molecular

[1]Ph.D. Program in Physics, The Graduate Center of the City University of New York, New York, NY 10016, USA. [2]Department of Physics, Brooklyn College of the City University of New York, Brooklyn, NY 11210, USA. [3]Ph.D. Program in Chemistry, The Graduate Center of the City University of New York, New York, NY 10016, USA. [4]Department of Chemistry, Lehman College of the City University of New York, Bronx, NY 10468, USA. ✉e-mail: yzhou4@gradcenter.cuny.edu; ngiovambattista@brooklyn.cuny.edu

dynamics simulations of water suggest that (ii') the PEL formalism may be used to predict the thermodynamic properties of liquids at low temperature in the presence of NQE[13].

In this work, we build upon the work presented in refs. 13,41,42 and extend the definition of configurational entropy to the case of quantum liquids. The configurational entropy $S_{IS}$[26-28], as well as the associated Kauzmann temperature[5], play important roles in the description of liquids and glasses in general[2,28,43-45], and are essential to predict the thermodynamic behavior of low-temperature liquids, including the corresponding equation of state. To do so, in this work, we hypothesize that the local minima (inherent structures, IS) in the PEL of the quantum liquid are isomorphic to the IS in the PEL of the corresponding classical liquid counterpart. This hypothesis, based on path-integral computer simulations of various model liquids, establishes a direct link between the configurational entropies of the quantum and associated classical liquid. Using path-integral computer simulations of a model liquid, we test that the configurational entropy defined in this study is consistent with relationships required by the PEL formalism.

It has been found that, for many classical liquids, the configurational entropy controls the corresponding liquid's dynamics. Specifically, classical molecular dynamics (MD) simulations of different liquids, including Lennard-Jones binary mixtures (LJBM)[43], silica[32], and water[40,46,47], show that the diffusion coefficient of the atoms/molecules in the system depends on the configurational entropy as predicted by the Adam-Gibbs (AG) relation[6]. This implies that, at a given temperature, not only the thermodynamics but also the dynamics of the liquids are controlled by the PEL topography. In this study, we perform ring-polymer molecular dynamics (RPMD) simulations and show that the AG relation, with the configurational entropy introduced here, holds for the quantum model liquid studied.

## Results

### The potential energy landscape formalism for quantum liquids

The PEL formalism provides a compact expression for the canonical partition function of the classical/quantum system of interest, $Q(N, V, T)$. Next, we present a brief overview of the PEL formalism for a quantum liquid composed of $N$ atoms (not necessarily identical but with a fixed composition); the generalization to molecular liquids is straightforward (see, e.g., ref. 13).

*Defining a PEL for a quantum liquid.* For a quantum system with fixed composition (e.g., a LJBM with a fixed ratio of particle numbers, $N_A/N_B$), the canonical partition function is given by

$$Q(N, V, T) = \text{Tr}(\hat{\rho}) \tag{1}$$

where $\text{Tr}(\hat{\rho})$ is the trace of the density operator $\hat{\rho} = \exp(-\beta\hat{H})$ and $\hat{H}$ is the Hamiltonian operator of the system,

$$\hat{H} = \sum_{i=1}^{N} \frac{\hat{\mathbf{p}}_i^2}{2m_i} + U(\hat{\mathbf{r}}_1, \hat{\mathbf{r}}_2, \ldots, \hat{\mathbf{r}}_N) \tag{2}$$

In Eq. (2), $U(\hat{\mathbf{r}}_1, \hat{\mathbf{r}}_2, \ldots, \hat{\mathbf{r}}_N)$ is the potential energy operator, and $(\hat{\mathbf{r}}_i, \hat{\mathbf{p}}_i)$ are the position and momentum operators associated with atom $i = 1, 2, \ldots, N$ ($\beta = 1/k_BT$ where $k_B$ is the Boltzmann's constant).

Using the path-integral formulation of quantum statistical mechanics, it can be shown that the canonical partition function of the quantum liquid (Eq. (1)) is mathematically identical to the canonical partition function of a classical system of $N$ (distinguishable) ring-polymers composed of $n_b \to \infty$ (distinguishable) beads[48,49].

Specifically,

$$Q(N, V, T) = \lim_{n_b \to \infty} \frac{1}{h^{3n_bN}} \int_V \left( \prod_{i=1}^{N} d\mathbf{r}_i^1 \cdots d\mathbf{r}_i^{n_b} \right) \int_{-\infty}^{\infty} \left( \prod_{i=1}^{N} d\mathbf{p}_i^1 \cdots d\mathbf{p}_i^{n_b} \right)$$
$$\exp(-\beta\mathcal{H}_{RP}(\mathbf{R}, \mathbf{P})) \tag{3}$$

where

$$\mathcal{H}_{RP}(\mathbf{R}, \mathbf{P}) = \sum_{i=1}^{N} \sum_{k=1}^{n_b} \frac{(\mathbf{p}_i^k)^2}{2m_i'} + \sum_{i=1}^{N} \sum_{k=1}^{n_b} \frac{1}{2} k_i^{sp}(\mathbf{r}_i^{k+1} - \mathbf{r}_i^k)^2 + \frac{1}{n_b} \sum_{k=1}^{n_b} U(\mathbf{r}_1^k, \mathbf{r}_2^k \ldots, \mathbf{r}_N^k) \tag{4}$$

is the Hamiltonian of the classical ring-polymer system. For each atom $i = 1, 2, \ldots, N$ of the quantum liquid with mass $m_i$, there is one and only one ring-polymer $i$ in the ring-polymer system associated to it. The spring constant of the corresponding ring-polymer is given by $k_i^{sp} = \frac{m_i n_b}{(\hbar\beta)^2}$ and the mass of the corresponding beads are given by $m_i' = n_b m_i$. In Eq. (3), $(\mathbf{r}_i^k, \mathbf{p}_i^k)$ are the vector position and momentum of the $k$-th bead of the $i$-th ring-polymer ($\mathbf{R} = \{\mathbf{r}_i^k\}$ and $\mathbf{P} = \{\mathbf{p}_i^k\}$; $i = 1, 2, \ldots, N$, $k = 1, 2, \ldots, n_b$). In Eq. (4), and throughout this work, $\mathbf{r}_i^1 = \mathbf{r}_i^{n_b+1}$ for $i = 1, 2, \ldots, N$ since the polymers are ring-polymers.

Eq. (4) implies that the potential energy of the ring-polymer system is given by

$$\mathcal{U}_{RP}(\mathbf{R}) = \sum_{i=1}^{N} \sum_{k=1}^{n_b} \frac{1}{2} k_i^{sp}(\mathbf{r}_i^{k+1} - \mathbf{r}_i^k)^2 + \frac{1}{n_b} \sum_{k=1}^{n_b} U(\mathbf{r}_1^k, \mathbf{r}_2^k \ldots, \mathbf{r}_N^k). \tag{5}$$

As explained in refs. 41,42, the function $\mathcal{U}_{RP}(\mathbf{R})$ defines a PEL that can be associated to the given quantum liquid. By applying the PEL formalism (originally proposed to study classical liquids[26]) to the PEL defined by Eq. (5), one can study the behavior of quantum liquids and, in particular, extract the corresponding thermodynamic properties[13,42]. We note that, strictly speaking, the above expressions hold for $n_b \to \infty$. However, in path-integral computational studies, one chooses a sufficiently large value of $n_b$ for which the thermodynamic properties of the system of interest converge (i.e., they no longer vary upon further increase in $n_b$).

The PEL formalism provides a simple understanding for the behavior of low-temperature liquids and glasses. Specifically, within the PEL formalism, a classical/quantum liquid is represented by a point on the PEL that moves over time (in the case of the quantum liquid, such a PEL is given by Eq. (5) with a fixed value of $n_b$). At high temperatures, the liquid has sufficient kinetic energy to overcome the potential energy barriers of the PEL and hence, it can explore different PEL basins. In the liquid state, the molecules/atoms are able to diffuse with time and the system moves on the PEL describing a trajectory. Instead, in the glass state, the molecules/atoms of the system are only able to vibrate about fixed positions. Accordingly, the representative point of the system in the PEL is limited to move within a single basin, about the corresponding local minimum or inherent structure (IS). While this picture holds for classical liquids and quantum liquids/ring-polymer systems, important differences exist in the corresponding PEL formalism. Among them is the fact that the PEL of a quantum liquid is $T$-dependent (since the spring constant $k_i^{sp} \propto T^2$) while the PEL of a classical system is not.

*Partition function in the PEL formalism.* As explained in detail in refs. 13,42, in the PEL formalism, the partition function of a quantum liquid (Eqs. (1) and (3)) can be written as,

$$Q(N, V, T) = \sum_{e_{IS}} e^{-\beta(e_{IS} - TS_{IS}(N, V, T, e_{IS}) + F_{vib}(N, V, T, e_{IS}))} \tag{6}$$

where $S_{IS}(N, V, T, e_{IS})$ is the configurational entropy of the system and $F_{vib}(N, V, T, e_{IS})$ is the corresponding vibrational Helmholtz free energy. Both quantities are precisely defined in the PEL formalism; see refs. 13,42. $S_{IS}(N, V, T, e_{IS})$ quantifies the number of IS available in the PEL with a given energy $e_{IS}$. $F_{vib}(N, V, T, e_{IS})$ is the contribution to the Helmholtz free energy of the system, $F(N, V, T)$, due to the explorations (by the system) of the PEL basins with IS energy $e_{IS}$.

The sum in Eq. (6) runs over all IS energies $e_{IS}$ available in the PEL. In the thermodynamic limit, one may employ the saddle point approximation[28], so that only the term that maximizes the sum in Eq. (6) dominates, i.e.,

$$Q(N, V, T) \approx e^{-\beta(E_{IS} - TS_{IS}(N, V, T, E_{IS}) + F_{vib}(N, V, T, E_{IS}))} \quad (7)$$

where $E_{IS}(N, V, T)$ is the solution to the following equation,

$$1 - T\left(\frac{\partial S_{IS}(N, V, T, e_{IS})}{\partial e_{IS}}\right)_{N, V, T} + \left(\frac{\partial F_{vib}(N, V, T, e_{IS})}{\partial e_{IS}}\right)_{N, V, T} = 0 \quad (8)$$

In computational studies, one identifies $E_{IS}(N, V, T)$ with the average value of $e_{IS}$ sampled by the system at the given working conditions $(N, V, T)$.

**Gaussian and harmonic PEL.** A common approximation in PEL studies is to assume that the PEL is Gaussian[27,28,43], i.e., that the distribution of IS energies $e_{IS}$ available in the PEL is given by a Gaussian distribution

$$\Omega_{IS}(N, V, T, e_{IS}) \approx \frac{1}{\sqrt{2\pi}\sigma} e^{\alpha N} e^{-(e_{IS} - E_0)^2/2\sigma^2} \quad (9)$$

This implies that the configurational entropy of the system is given by

$$S_{IS}(N, V, T, e_{IS}) \approx k_B\left[\alpha N - \frac{(e_{IS} - E_0)^2}{2\sigma^2}\right] \quad (10)$$

Here, $\alpha$, $E_0$, and $\sigma$ are PEL variables that depend, in principle, on $(V, T)$. Importantly, MD/PIMD simulations of very different liquids, including LJBM[43,50], water[13,39,46], ortho-terphenyl[37], and water-like monatomic systems[42,51], indicate that the corresponding PEL is Gaussian.

Another common approximation in the PEL formalism is to assume that the PEL basins are quadratic functions (of the atoms/ring-polymer beads coordinates) near the corresponding IS. Under the harmonic approximation of the PEL, one can show that

$$F_{vib}(N, V, T, e_{IS}) \approx F_{vib}^{harm}(N, V, T, e_{IS}) = 3Nn_b k_B T\ln(\beta\hbar\omega_0) + k_B T\mathcal{S}(N, V, T, e_{IS}) \quad (11)$$

where

$$\mathcal{S}(N, V, T, e_{IS}) \approx \left\langle \ln\left(\prod_{j=1}^{3n_b N}(\omega_j/\omega_0)\right)\right\rangle_{e_{IS}} \quad (12)$$

is the basin shape function. The $3n_b N$ values $\{\omega_j^2 = \omega_j^2(N, V, T, e_{IS})\}$ are the eigenvalues of the mass-weighted Hessian matrix of the ring-polymer system evaluated at the IS with energy $e_{IS}$; $< \ldots >_{e_{IS}}$ indicates an average over all basins of the PEL with energy $e_{IS}$. The constant $\omega_0$ is an arbitrary quantity that makes the argument of $\ln(\ldots)$ dimensionless. $\mathcal{S}(N, V, T, e_{IS})$ quantifies the average local curvature of the PEL basins with IS energy $e_{IS}$ and it is the only term in Eq. (11) that makes $F_{vib}$ dependent on the PEL of the system. We note that Eqs. (7)–(12) also apply to the case of classical liquids ($n_b = 1$). However, since the PEL of classical liquids is $T$-independent, the topographic properties of the PEL are also $T$-independent; specifically, $S_{IS} = S_{IS}(N, V, e_{IS})$, $\omega_j = \omega_j(N, V, e_{IS})$ and $\mathcal{S} = \mathcal{S}(N, V, e_{IS})$.

In the PEL formalism, the Helmholtz free energy of the system, $F(N, V, T) = -k_B T\ln(Q(N, V, T))$, follows directly from Eq. (7),

$$F(N, V, T) = E_{IS}(N, V, T) - TS_{IS}(N, V, T, E_{IS}) + F_{vib}(N, V, T, E_{IS}) \quad (13)$$

In the case of a Gaussian and harmonic PEL (Eqs. (10) and (11)), it can be shown from Eq. (13) that the free energy of the quantum liquid can be expressed in terms of only three PEL variables $\{\alpha, E_0, \sigma\}$, and the eigenvalues of the mass-weighted Hessian matrix of the corresponding ring-polymer system; see, e.g., refs. 27–29,42. Accordingly, one can obtain an analytical expression for all the thermodynamic properties of the quantum liquid, including the equation of state[36,39,40,51]. In particular, it can be shown that, for a Gaussian and harmonic PEL, the total energy of the system, $E(N, V, T) = (\partial(\beta F)/\partial\beta)_{N,V}$, is given by (see Sec. VI of the SI)

$$E(N, V, T) = E_{IS}(N, V, T) + E_{vib}(N, V, T) \quad (14)$$

where

$$E_{IS}(N, V, T) \approx E_{IS}^{harm}(N, V, T) = E_0 - \sigma^2(\beta + b) \quad (15)$$

and

$$E_{vib}(N, V, T) \approx E_{vib}^{harm}(N, V, T) = 3Nn_b k_B T + \left(\frac{\partial\mathcal{S}(N, V, T, E_{IS})}{\partial\beta}\right)_{N, V, E_{IS}} \quad (16)$$

In Eq. (15), $b \equiv \left(\frac{\partial \mathcal{S}(N, V, T, e_{IS})}{\partial e_{IS}}\right)_{N, V, T, e_{IS} = E_{IS}}$. As explained in ref. 42, to obtain Eqs. (15) and (16), we have assumed that the parameters $\alpha = \alpha(V)$, $E_0 = E_0(V)$, and $\sigma = \sigma(V)$, i.e., they are $T$-independent. We note that Eq. (15) and (16) are also valid for classical liquids. In the classical case, however, Eq. (16) reduces to $E_{vib} = 3NkT$ since $n_b = 1$ and the last term of Eq. (16) vanishes (the PEL for classical system is $T$-independent). Alternatively, it can be shown that within the harmonic approximation of the PEL, $\left(\frac{\partial \mathcal{S}(N, V, T, e_{IS})}{\partial\beta}\right)_{N, V, e_{IS} = E_{IS}} = -2E_{sp}(N, V, T)$ where $E_{sp}(N, V, T) \equiv \langle\sum_{i=1}^{N}\sum_{k=1}^{n_b}\frac{1}{2}k_i^{sp}(\mathbf{r}_i^{k+1} - \mathbf{r}_i^k)^2\rangle_{N, V, T}$ is the average potential energy of the ring-polymer springs [see Sec. VII of the SI]. For classical systems, $n_b = 1$ and hence, $E_{sp}(N, V, T) = 0$ (there are no springs associated to the liquid atoms).

Relevant to this work, we note that Eqs. (15) and (16) hold for the classical LJBM[52,53] suggesting that its PEL is indeed Gaussian and harmonic. Below we show that Eqs. (15) and (16) also hold for quantum LJBM if the Planck's constant is small (mild quantumness, $h = h_a$) but fail as the Planck's constant is increased. For large values of $h$, anharmonic correction to the harmonic approximation of the PEL are needed.

**Anharmonic corrections in the PEL formalism.** Anharmonic corrections to the PEL have been introduced in the past to study the equation of state of classical liquids using the PEL formalism[28,37,39,40]. In most studies, anharmonicities were assumed to depend only on $T$, implying that the basins anharmonicities were independent of where the basins were located within the PEL (the basins anharmonicities were assumed to be independent of the basin IS energy $e_{IS}$). Here, we present a general approach where the anharmonic corrections to the PEL may vary with both $T$ and $e_{IS}$.

In the presence of anharmonic corrections, one can formally express the vibrational Helmholtz free energy of the system as

$$F_{vib}(N, V, T, e_{IS}) = F_{vib}^{harm}(N, V, T, e_{IS}) + F_{vib}^{anh}(N, V, T, e_{IS}), \quad (17)$$

where $F_{vib}^{harm}(N, V, T, e_{IS})$ is given by Eq. (11) and $F_{vib}^{anh}(N, V, T, e_{IS})$ is the correction due to the basins anharmonicity. The formal expression for

the total energy of the system, $E(N, V, T) = (\partial(\beta F(N, V, T))/\partial\beta)_{N,V}$, follows from Eqs. (13) and (17). It can be shown that $E(N, V, T)$ is also given by Eq. (14), however, the new expressions for $E_{IS}$ and $E_{vib}$ (for a Gaussian PEL with anharmonic corrections; in equilibrium, $e_{IS} \to E_{IS}$) change as follows (see Sec. VI of the SI).

$$E_{IS}(N, V, T) = E_{IS}^{harm}(N, V, T) + E_{IS}^{anh}(N, V, T) \qquad (18)$$

where $E_{IS}^{harm}(N, V, T)$ is given by Eq. (15), and

$$E_{IS}^{anh}(N, V, T) = -\sigma^2 \left( \frac{\partial(\beta F_{vib}^{anh}(N, V, T, e_{IS}))}{\partial e_{IS}} \right)_{N, V, T, e_{IS} = E_{IS}}. \qquad (19)$$

The new expression for $E_{vib}$ is given by

$$E_{vib}(N, V, T) = E_{vib}^{harm}(N, V, T) + E_{vib}^{anh}(N, V, T, E_{IS}) \qquad (20)$$

where $E_{vib}^{harm}(N, V, T)$ is given by Eq. (16), and

$$E_{vib}^{anh}(N, V, T) = \left( \frac{\partial(\beta F_{vib}^{anh}(N, V, T, E_{IS}))}{\partial\beta} \right)_{N, V, e_{IS} = E_{IS}}. \qquad (21)$$

Eqs. (19) and (21) are formal expressions that yield $E_{IS}^{anh}$ and $E_{vib}^{anh}$ once $F_{vib}^{anh}$ is provided. As an example, we consider the case where the anharmonic corrections depend only on $T$. In this case, one can write the following general expression,[28]

$$F_{vib}^{anh}(N, V, T, e_{IS}) = \sum_{i=2}^{i_{max}} \frac{c_i}{1-i} T^i, \qquad (22)$$

where the coefficients $\{c_i\}$ depend only on $(N, V)$. In this case, Eq. (21) implies that

$$E_{vib}^{anh}(N, V, T) = \sum_{i=2}^{i_{max}} c_i T^i \qquad (23)$$

while, from Eq. (19), $E_{IS}^{anh} = 0$. Eqs. (22) and (23) are consistent with refs. 39,40.

In this work, we will model the anharmonicities of the PEL basins of the quantum LJBM as $\beta F_{vib}^{anh}(N, V, T, e_{IS}) = \sum_{i=0}^{\infty} \tilde{B}_i(N, V, T) e_{IS}^i$ where the coefficients $\tilde{B}_i(N, V, T)$ can be obtained from the RPMD simulations (see below). However, we find that keeping this expansion up to first order in $e_{IS}$ is sufficient to fit the properties of the LJBMs studied in this work,

$$\beta F_{vib}^{anh}(N, V, T, e_{IS}) = \tilde{B}_0(N, V, T) + \tilde{B}_1(N, V, T) e_{IS} \qquad (24)$$

Using Eqs. (19) and (21), one obtains

$$E_{IS}^{anh}(N, V, T) = -\sigma^2 \tilde{B}_1(N, V, T) \qquad (25)$$

and

$$E_{vib}^{anh}(N, V, T, E_{IS}) = \left( \frac{\partial\tilde{B}_0(N, V, T)}{\partial\beta} \right)_{N, V} + \left( \frac{\partial\tilde{B}_1(N, V, T)}{\partial\beta} \right)_{N, V} E_{IS}(N, V, T) \qquad (26)$$

To summarize, for a Gaussian PEL with anharmonicities given by Eq. (24), the following expressions hold (using Eqs. (17), (11), (24) for $F_{vib}$; Eqs. (18), (15), (25) for $E_{IS}$; and Eqs. (20), (16), (26) for $E_{vib}$),

$$F_{vib}(N, V, T, e_{IS}) = 3Nn_b k_B T \ln(\beta\hbar\omega_0) + k_B T \mathcal{S}(N, V, T, e_{IS}) + k_B T \tilde{B}_0(N, V, T) + k_B T \tilde{B}_1(N, V, T) e_{IS} \qquad (27)$$

$$E_{IS}(N, V, T) = E_0(V) - \sigma(V)^2 (\beta + b(N, V, T) + \tilde{B}_1(N, V, T)) \qquad (28)$$

$$E_{vib}(N, V, T) = 3Nn_b k_B T + \left( \frac{\partial\mathcal{S}(N, V, T, E_{IS})}{\partial\beta} \right)_{N, V, E_{IS}} + \left( \frac{\partial\tilde{B}_0(N, V, T)}{\partial\beta} \right)_{N, V} + \left( \frac{\partial\tilde{B}_1(N, V, T)}{\partial\beta} \right)_{N, V} E_{IS}(N, V, T) \qquad (29)$$

**Hypotheses and protocols.** The theoretical predictions of the PEL formalism discussed in the previous section are based on a handful of PEL variables, $\{\alpha, E_0, \sigma^2, b, \mathcal{S}, \tilde{B}_0, \tilde{B}_1\}$. For any practical purpose, these quantities need to be fit to properties obtained from the RPMD simulations. Next, we explain how these quantities are obtained, and the approximations involved.

*Quantities* $\{\mathcal{S}, b\}$. For the classical LJBM, the normal mode frequencies $\{\omega_{j,0}\}$ with $j = 1, 2, ..., 3N$ are calculated by diagonalizing the corresponding mass-weighted Hessian matrix evaluated at the IS sampled by the system at the given $(N, V, T)$; the eigenvalues of the mass-weighted Hessian matrix are $\{(\omega_{j,0})^2\}$. For the quantum liquid/ring-polymer system, there are $3n_b N$ normal mode frequencies $\{\omega_j\}$ ($j = 1, 2, ..., 3n_b N$) given by the eigenvalues of the mass-weighted Hessian matrix of the ring-polymer system evaluated at the corresponding IS [Eq. (12)]. In this case, we follow the procedure described in refs. 13,42 to obtain the ring-polymer (IS) normal mode frequencies analytically, using the normal mode frequencies $\{\omega_{j,0}\}_{j=1,2,...,3N}$ of the corresponding classical system; see Eq. 20 in ref. 13. The obtained normal mode frequencies are then used in Eq. (12) to calculate the shape function of the system. As shown in Sec. I of the SI, and consistent with previous MD/PIMD studies of classical and quantum liquids, our path-integral computer simulations of the LJBM show that

$$\mathcal{S}(N, V, T, e_{IS}) \approx a(N, V, T) + b(N, V, T) e_{IS} \qquad (30)$$

for all values of $h$ studied. Note that for quantum liquids, the variables $a$ and $b$ depend on $(N, V, T)$ while, in the classical case, they depend only on $(N, V)$.

*Quantities* $\{\alpha, E_0, \sigma^2\}$. These parameters characterize the configurational entropy of the system $S_{IS}(N, V, T, e_{IS})$ and hence, they specify how the IS are distributed within the PEL (Eqs. (9) and (10)). In our previous studies[13,42], we assumed that, for the quantum liquids, $\{\alpha, E_0, \sigma^2\}$ were independent of $T$. This implies that, while the PEL of the quantum liquid is $T$-dependent, the distribution of IS within the PEL is not.

In previous studies[13,41,42], we found that the Gaussian approximation of the PEL works remarkably well for atomistic and molecular quantum liquids. In the absence of anharmonicities, Eq. (15) was found to be in very good agreement with the values of $E_{IS}(T)$ obtained from PI computer simulations[13,42]. Importantly, (i) it was found that the parameters $\{\alpha, E_0, \sigma^2\}$ (in the absence of anharmonicities) depend on the quantumness of the liquid (i.e., they all depend on the Planck's constant $h$). This is not unreasonable since, varying $h$, changes the Hamiltonian of the ring-polymer system (Eq. (4)). However, we also found indications that the values of $\{\alpha, E_0, \sigma^2\}$ should not depend on the nature (classical/quantum) of the liquid (as quantified by $h$). Specifically, if the values of $\{\alpha, E_0, \sigma^2\}$ vary with $h$ then the distributions of the IS in the PEL of classical and quantum liquids must be different. For this to be the case, there should be IS in the ring-polymer system PEL (RP-PEL) where the ring-polymers are not collapsed – as explained in ref. 42, if the ring-polymers are collapsed at the IS of the RP-PEL then such IS (in the RP-PEL) also define IS of the classical liquid PEL (CL-PEL), and vice versa[42]. However, (ii) for all the quantum liquids studied[13,41,42], it was always found that the ring-polymers were collapsed at the IS of the RP-PEL sampled by the system. There are two options for the findings (i) and (ii)

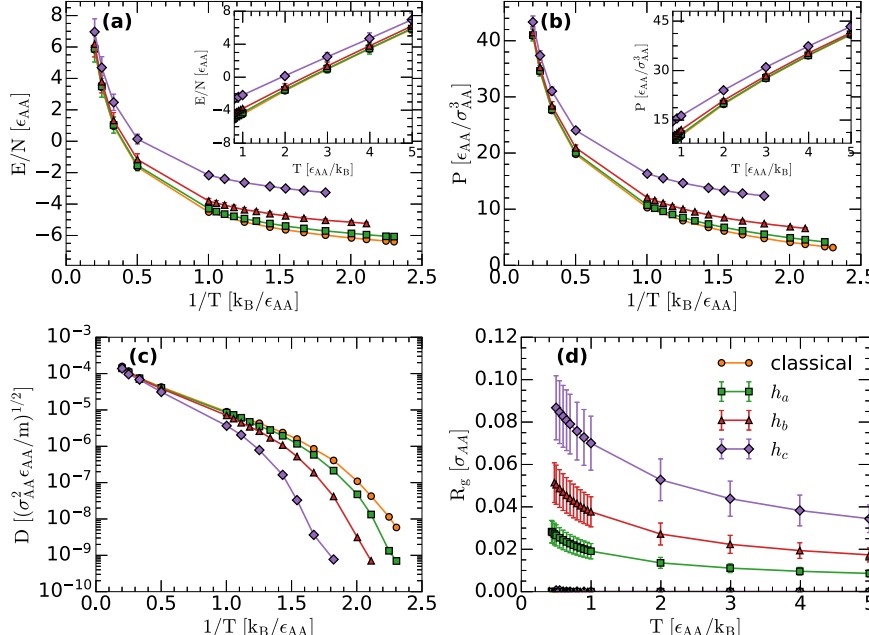

**Fig. 1 | Thermodynamic and dynamic properties of the classical and quantum LJBMs from MD and RPMD simulations. a** Total energy and (**b**) pressure of the LJBMs as a function of temperature ($\rho = 1.2$) obtained from classical MD and RPMD simulations with Planck's constant $h_a = 0.2474$, $h_b = 0.5000$, and $h_c = 1.0000$. As $h$ increases, and the LJBMs become more quantum, both $E(T)$ and $P(T)$ increase, particularly at low temperatures. **c** Diffusion coefficient $D(T)$ of the classical and quantum LJBMs studied (A-type atoms). Consistent with ref. 54, at a given low temperature, $D(T)$ decreases with increasing quantumness, as quantified by $h$, for

the range of $h$-values studied. **d** Radius of gyration $R_g$ of the A-type atoms of the LJBMs studied (solid symbols). In all cases, $R_g(T)$ increases upon cooling. The observed atoms delocalization becomes more pronounced with increasing $h$. The empty symbols in (**d**) correspond to the values of $R_g$ at the IS sampled by the system; $R_g(T) \approx 0$ at the IS implying that the ring-polymers are collapsed. Error bars in (**d**) represent the standard deviation. Source data are provided as a Source Data file.

to be compatible. (A) One may assume that the quantities $\{\alpha, E_0, \sigma^2\}$ vary with the quantumness of the liquid but the IS of the RP-PEL with non-collapsed ring-polymers are rare, difficult to sample in the PI computer simulations. This is the approach followed originally in ref. 42. Alternatively, one may consider that (B) there are no IS in the RP-PEL where ring-polymers are not collapsed at the working conditions studied and hence, the quantities $\{\alpha, E_0, \sigma^2\}$ are independent of whether the liquid obeys classical or quantum mechanics. In this work, we will depart from the approach followed in ref. 42 and assume that option (B) holds. Accordingly, the values of $\{\alpha, E_0, \sigma^2\}$ are considered to be $h$-independent and hence, they will be extracted from the classical MD simulations following the same procedure from previous classical MD simulations[39,40,43,46,53]. It follows from Eq. (10), that hypothesis (B) also implies that the configurational entropy $S_{IS}$ of the system as a function of $e_{IS}$ is identical for the classical and quantum liquid. As we will show, in the scenario (B), anharmonicities may need to be included.

*Quantities* $\{\widetilde{B}_0, \widetilde{B}_1\}$. As a general expression, we will consider that, for a fixed value of $(N, V)$,

$$\widetilde{B}_j(T) = \sum_{i=0}^{i_{max}} c_{j,i} T^i \tag{31}$$

where $j = 0, 1$. We find that a value of $i_{max} = 2$ is sufficient to reproduce the results from the RPMD simulations. As will be shown below, the coefficients $c_{j=1,i}$ ($i = 0, 1, 2$) can be obtained from the RPMD simulations using Eq. (25). Similarly, the coefficients $c_{j=0,i}$ for $i = 1, 2$ can be obtained from the RPMD simulations using Eq. (26). The coefficient $c_{0,0}$ cannot be obtained from Eq. (26) and is evaluated from the configurational entropy $S_{IS}$, as explained below.

In order to validate hypothesis (B), in Sec. 2.2 we will test whether the configurational entropy given in Eq. (10), with the parameters $\{\alpha, E_0, \sigma^2\}$ obtained from classical simulations, also hold for the quantum LJBMs studied. To do so, we note that the probability for the

system to sample basins with IS energy $e_{IS}$, at a given $T$ ($N, V$ are constant) is given by (for a simpler notation, as in Eq. (6), we assume that the values of $e_{IS}$ are discrete)

$$P(T, e_{IS}) = \frac{e^{-\beta(e_{IS} - TS_{IS}(N, V, T, e_{IS}) + F_{vib}(N, V, T, e_{IS}))}}{Q(N, V, T)} \tag{32}$$

Accordingly, $S_{IS}(N, V, T, e_{IS})$ must obey the following expression (see Eqs. (11) and (17)),

$$S_{IS}(N, V, T, e_{IS})/k_B = \ln(P(T, e_{IS})) + 3Nn_b \ln(\beta \hbar \omega_0)$$
$$+ S(N, V, T, e_{IS}) + \beta e_{IS} + \beta F_{vib}^{anh}(N, V, T, e_{IS}) + \ln(Q(N, V, T)) \tag{33}$$

In this work, we will calculate $P(T, e_{IS})$ directly from the RPMD simulations of the target LJBMs and use Eq. (33) to validate the expression for $S_{IS}(N, V, T, e_{IS})$ obtained from the PEL formalism (with the hypothesis (B) discussed above), Eq. (10).

## Potential energy landscape of classical and quantum Lennard-Jones binary mixtures

**Equilibrium properties.** We first focus on the total energy $E(T)$, pressure $P(T)$, and diffusion coefficient $D(T)$ (A-type atoms) of the LJBMs with different levels of quantumness. Figure 1a–c show $E(T)$, $P(T)$, and $D(T)$ for the LJBMs obtained from classical MD and RPMD simulations with $h = h_a, h_b, h_c$. At high temperatures, the values of $E(T)$, $P(T)$, and $D(T)$ obtained for different values of $h$ become closer and closer with increasing temperatures. Hence, quantum effects do not affect the thermodynamic and dynamical properties of the LJBMs at very high temperatures (as expected). Instead, at low temperatures ($T < 1.0$), both $E(T)$ and $P(T)$ increase with increasing $h$, as the LJBMs become more quantum. Consistent with refs. 24,54,55, at a given low temperature, $D(T)$ decreases with increasing values of $h$, for the range of

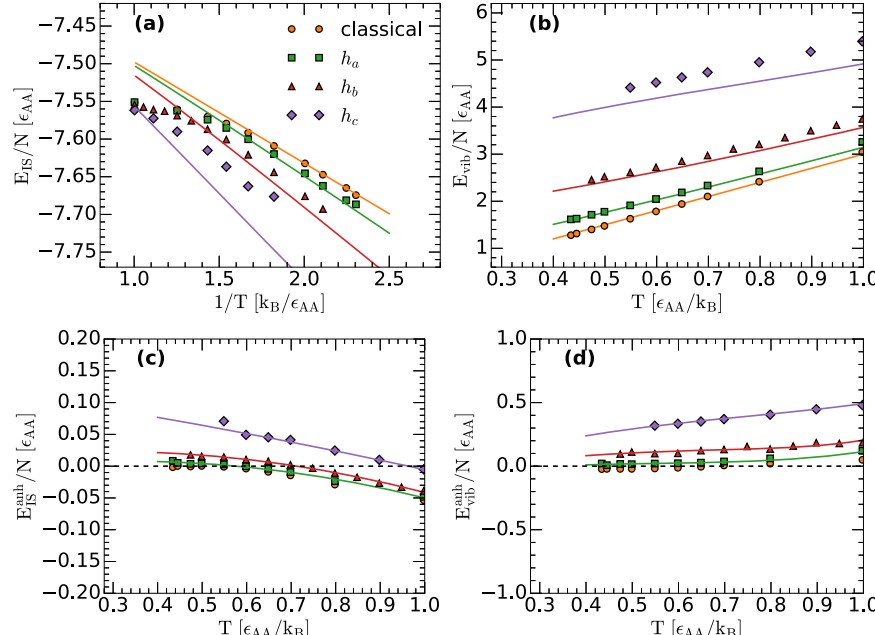

**Fig. 2 | Harmonic and anharmonic contributions to the inherent structure and vibrational energies. a** Inherent structure energy $E_{IS}(T)$ and (**b**) vibrational energy $E_{vib}(T)$ as a function of temperature for the LJBMs studied. Results are obtained from classical MD and RPMD simulations with $h = h_a, h_b, h_c$ (symbols). Lines are the predictions of the PEL formalism for a Gaussian and harmonic PEL, $E_{IS}^{harm}(T)$ and $E_{vib}^{harm}(T)$, given in Eqs. (15) and (16). **c** Anharmonic contributions to the IS energy, $E_{IS}^{anh}(T) = E_{IS}(T) - E_{IS}^{harm}(T)$, obtained from (**a**) [symbols]. Lines are the fit to the data points using Eq. (25) and (31). **d** Anharmonic contributions to the vibrational energy, $E_{vib}^{anh}(T) = E_{vib}(T) - E_{vib}^{harm}(T)$, obtained from (**b**) [symbols]. Lines are the fit to the data points using Eq. (26) and (31). Anharmonicities are negligible for the classical and weakly-quantum LJBMs ($h = h_a$), but they become relevant as the quantumness of the LJBM increases ($h = h_b, h_c$). Dashed-lines in (c)(d) correspond to zero energy. Source data are provided as a Source Data file.

$h$-values studied, which is counterintuitive since NQE are expected to speed-up the liquid dynamics. The NQE at low temperatures are due to the quantum delocalization of the atoms $h$. Indeed, as shown in Fig. 1d, the radius of gyration $R_g(T)$ of the ring-polymers associated to the LJBMs (A-type atoms) increases with increasing $h$. We note that the atoms delocalization in the LJBMs studied is mild but not negligible. At the lowest temperatures studied, $R_g \lesssim 0.09$; a value of $R_g = 0.09$ indicates that the ring-polymers expand over a sphere of radius approximately equal to 9% of the A-type atom hard-core radius.

**PEL of the LJBMs.** The IS and vibrational energy of the studied LJBMs, $E_{IS}(T)$ and $E_{vib}(T)$, are included in Fig. 2a, b. The results from the RPMD simulations are indicated by symbols; the lines are the prediction using Eqs. (15) and (16) for the case where the PEL of the classical/quantum LJBMs are assumed to be Gaussian and harmonic. As explained above, we consider that the distribution of IS energies in the CL-PEL and RP-PEL are identical, i.e., in both cases, $\Omega_{IS}$ is given by Eq. (9) with the same values of $\{\alpha, E_0, \sigma^2\}$. It follows from Fig. 2a, b that, when nuclear quantum effects are neglected (MD simulations) or small (PIMD simulations with $h = h_a$), the Gaussian and harmonic approximation of the PEL are fully consistent with the corresponding simulations of the LJBMs (up to relatively high temperatures, approximately $T \lesssim 0.8$). However, as the quantum nature of the LJBM increases ($h = h_b, h_c$), deviations between the predictions of the PEL formalism (within the Gaussian and harmonic approximation of the PEL, Eqs. (15) and (16)) and RPMD simulations become evident in both $E_{IS}(T)$ and $E_{vib}(T)$. Accordingly, for $h = h_b, h_c$, anharmonic corrections are needed.

The anharmonic corrections to $E_{IS}(T)$ and $E_{vib}(T)$ can be calculated numerically from Fig. 2a, b and using the expressions $E_{IS}^{anh}(T) = E_{IS}(T) - E_{IS}^{harm}(T)$ and $E_{vib}^{anh}(T) = E_{vib}(T) - E_{vib}^{harm}(T)$. The corresponding values of $E_{IS}^{anh}(T)$ and $E_{vib}^{anh}(T)$ are indicated by symbols in Fig. 2c, d. Also included in Fig. 2c, d are the fits to $E_{IS}^{anh}(T)$ and $E_{vib}^{anh}(T)$ using Eqs. (25) and (26), respectively, and Eq. (31) (lines). The fits to the data points in Fig. 2c, d

are excellent indicating that, for the LJBMs studied, the anharmonic contributions to the vibrational free energy can indeed be modeled using Eq. (24) with the coefficients $\widetilde{B}_j$ given by Eq. (31) ($\widetilde{B}_0$ and $\widetilde{B}_1$ are shown in Fig. S9 of SI).

For comparison, we include in Fig. 3 the values of $E_{IS}(T)$ and $E_{vib}(T)$ obtained numerically (taken from Fig. 2a, b); symbols) together with the corresponding predictions of the PEL formalism including the anharmonic corrections; see Eqs. (28) and (29). The agreement between the predictions of the PEL formalism (with the Gaussian approximation and including anharmonicities) and RPMD simulations is excellent. Importantly, Fig. 3 supports that hypothesis (B) indeed holds for the LJBMs studied.

**Configurational Entropy.** Next, we study the configurational entropy of the quantum LJBMs. We also test whether the hypotheses of this work are self-consistent within the PEL formalism. Specifically, our results are based on the hypotheses that (i) $\Omega_{IS}(e_{IS})$ is identical for the classical and quantum liquids (given by the Gaussian approximation, Eq. (9), with identical PEL variables $\{\alpha, E_0, \sigma^2\}$), and that (ii) the PEL anharmonicities can be modeled using Eq. (24).

Hypothesis (i) implies that for all the LJBMs studied, $S_{IS}(N, V, T, e_{IS}) = S_{IS}(N, V, e_{IS})$, independent of $h$, with $S_{IS}(N, V, e_{IS})$ given by Eq. (10). Again, this implies that, at a given depth $e_{IS}$ of the corresponding PEL, the classical and quantum liquids have access to the same number of IS [Fig. 4a shows the $S_{IS}(N, V, e_{IS})$ of the LJBMs as a function of $e_{IS}$]. However, the number of IS accessible to the LJBMs at a given temperature does depend on the quantumness of the LJBM considered. To show this, we include in Fig. 4b, the $S_{IS}$ of the quantum LJBMs as a function of $T$, after substituting $E_{IS}(T)$ using Eq. (28). Since $E_{IS}(T)$ varies with $h$ (Fig. 3), the temperature-dependence of $S_{IS}$ is different for the quantum LJBMs studied. Our RPMD simulations show that $S_{IS}(T)$ shifts towards higher temperatures as $h$ increases. In particular, the Kauzmann temperature $T_K$, defined as the temperature at

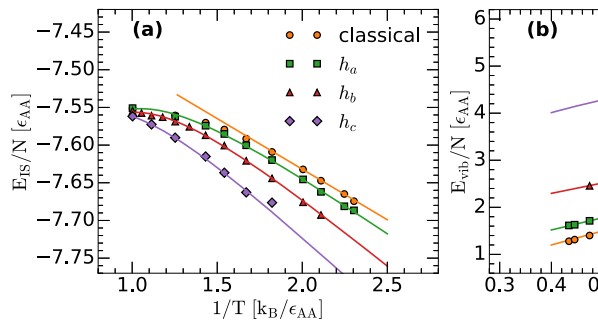

**Fig. 3 | Inherent structure and vibrational energies from computer simulations and PEL formalism. a** Inherent structure energy $E_{IS}(T)$ and (**b**) vibrational energy $E_{vib}(T)$ as a function of temperature for the LJBMs studied. Results are obtained from RPMD simulations [symbols; taken from Fig. 2a, b]. Lines are the predictions of the PEL formalism for a Gaussian and anharmonic PEL, using Eqs. (28) and (29) [see text and Fig. 2c, d]. Source data are provided as a Source Data file.

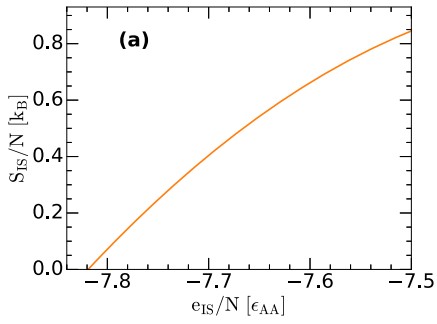

**Fig. 4 | Configurational entropy of the classical and quantum LJBMs.**
**a** Configurational entropy as a function of the IS energy, $S_{IS}(e_{IS})$, for the classical and quantum LJBMs studied [Eq. (10)]; $S_{IS}(e_{IS})$ is considered to be independent of the quantum character of the LJBM considered (see text). $S_{IS}(e_{IS})$ is obtained by thermodynamic integration and classical MD simulations of the LJBM, following the procedure of refs. 43,53 (see Sec. III and Sec. IV of the SI). **b** Configurational entropy as a function of temperature, $S_{IS}(T)$. The dashed line is the $S_{IS}(T)$ for the classical LJBM (see Sec. III and Sec. IV of the SI). The $S_{IS}(T)$ for the quantum LJBMs (solid lines) are obtained from (**a**) [Eq. (10)] and substituting $e_{IS} \rightarrow E_{IS}$ using Eq. (28). The symbols are the values of $S_{IS}(T)$ obtained numerically from PI computer simulations and thermodynamic integration, as explained in Sec. III and Sec. IV of the SI. The numerical results for $S_{IS}(T)$ (symbols) are in very good agreement with the corresponding values predicted by the PEL formalism (solid lines). Source data are provided as a Source Data file.

which $S_{IS}(T) = 0$, increases considerably as the liquid becomes more quantum. Specifically, $T_K = 0.28, 0.32, 0.35, 0.40$ for classical, $h_a$, $h_b$, $h_c$, respectively. This result is unexpected, since it suggests that the glass transition temperature increases as the system becomes more quantum. While surprising, our results are fully consistent with the work of Markland et al.[24,54] who find that the diffusivity of atoms in the LJBM studied here is suppressed as the system becomes more quantum [for low and intermediate values of $h$, comparable to the values studied in this work]. Overall, our results imply that, at a given temperature, the LJBMs explore IS located at different depths of the corresponding PEL (Fig. 3) and hence, they have access to a different number of IS in the corresponding PEL (Fig. 4b).

To validate the results in Fig. 4b, we also evaluate $S_{IS}(T)$ numerically, by combining PI computer simulations and thermodynamic integration. This procedure to evaluate $S_{IS}(T)$ is based on the definition of $S_{IS}$ within the PEL formalism, $S_{IS} = S - S_{vib}$, where $S_{vib}$ is the vibrational entropy of the system; this procedure has been implemented in the past to calculate the $S_{IS}(T)$ of classical liquids, including atomistic model liquids and water[39,40,43,53]. The details of these calculations are included in Secs. III and IV of the SI. The numerical values of $S_{IS}(T)$ for the quantum LJBMs studied are indicated by the symbols in Fig. 4b, and are in very good agreement with the predictions of the PEL (lines).

To further confirm that the calculated configurational entropy of the LJBMs (Fig. 4) are physically sound (i.e., self-consistent with the PEL formalism), we test that our results are consistent with Eq. (33). A similar self-consistency test was implemented for the case of the classical LJBM in refs. 43,52,53. Eq. (33) imposes a strict relationship

between $S_{IS}(N, V, e_{IS})$ and the probability distribution $P(T, e_{IS})$. Specifically, at fixed $(N, V)$, the PEL formalism (Eq. (33)) requires that

$$
\begin{aligned}
S_{IS}(e_{IS})/k_B = \ln(P(T, e_{IS})) + 3Nn_b\ln(\beta\hbar\omega_0) \\
+ \mathcal{S}(T, e_{IS}) + \beta e_{IS} + \beta F_{vib}^{anh}(T, e_{IS}) + c(T)
\end{aligned}
\tag{34}
$$

where $c(T)$ is a quantity independent of $e_{IS}$. Accordingly, independently of the temperature considered, the right-hand-side of Eq. (34) should overlap with $S_{IS}(e_{IS})$ (up to some unknown function $c(T)$). We note that $P(T, e_{IS})$ is calculated numerically from the RPMD simulations and hence, Eq. (34) provides a strong test to the validity of the $S_{IS}(N, V, e_{IS})$ reported in this work as well as to the underlying hypotheses (i) and (ii) stated above. Figure 5 shows the $S_{IS}(N, V, e_{IS})$ of the quantum LJBMs for $h = h_a, h_b, h_c$ [black line; taken from Fig. 4a] together with the corresponding fit given by the right-hand-side of Eq. (34) (the value of $c(T)$ is adjusted for a maximum overlap between both sides of Eq. (34))[53]. It follows from Fig. 5 that Eq. (34) holds remarkably well up to approximately $T = 1.0$, which is a rather high temperature for the LJBMs studied. Additional tests regarding the importance of the anharmonic contributions to $S_{IS}$ are included in Sec. II of the SI.

**Adam-Gibbs relation.** The Adam-Gibbs relation relates the diffusion coefficient $D(T)$ of a liquid with its configurational entropy as follows[6],

$$
D(T) = D_0 \exp\left(-\frac{A}{TS_{IS}}\right)
\tag{35}
$$

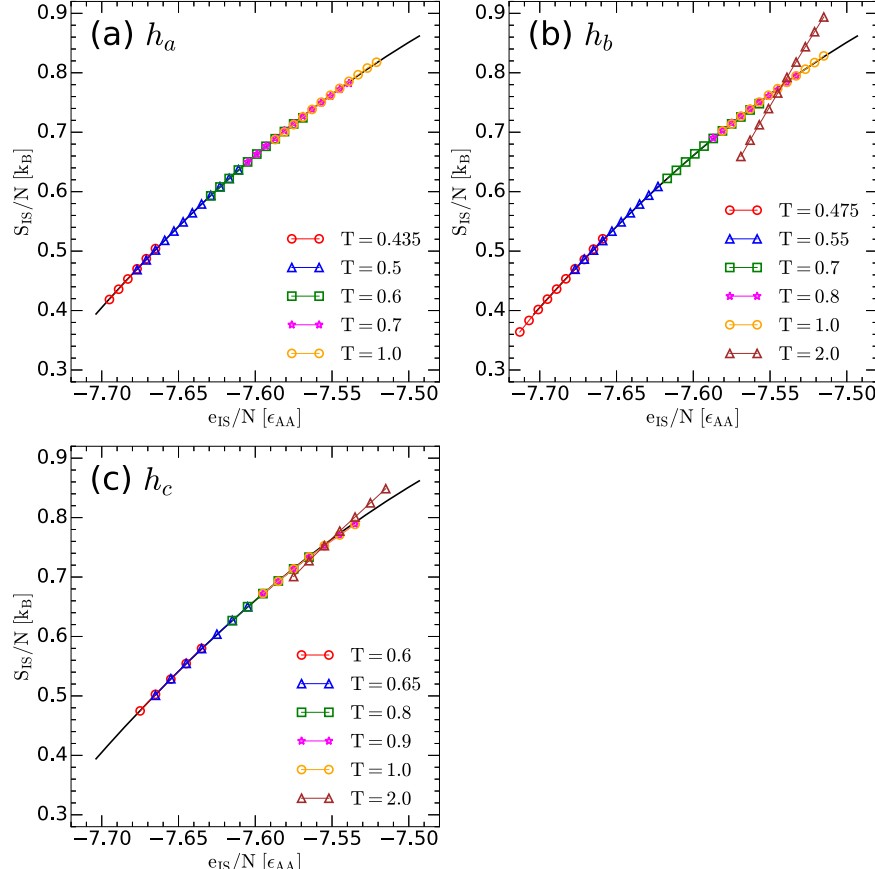

**Fig. 5 | Validation of the configurational entropy for the classical and quantum LJBMs. a** Configurational entropy of the LJBMs studied as a function of the IS energy $e_{IS}$, $S_{IS}(e_{IS})$ [solid black line; taken from Fig. 4(a)]. The symbols correspond to the expression on the right-hand-side of Eq. (34) for temperatures $T = 0.435$ (red circles), 0.5 (blue up-triangles), 0.6 (green squares), 0.7 (magenta stars) and 1.0 (orange circles). Results at the different temperatures are obtained from RPMD simulations with $h = h_a$ [datasets are shifted by a constant $c(T)$; see text]. **b** Same as (**a**) for $h = h_b$, and for temperatures $T = 0.475$ (red circles), 0.55 (blue up-triangles), 0.7 (green squares), 0.8 (magenta stars), 1.0 (orange circles), and 2.0 (maroon up-triangles). **c** Same as (**a**) for $h = h_c$, and for temperatures $T = 0.6$ (red circles), 0.65 (blue up-triangles), 0.8 (green squares), 0.9 (magenta stars), 1.0 (orange circles), and 2.0 (maroon up-triangles). In all cases, the symbols fully overlap with the black line at $T \leq 1.0$, implying that Eq. (34) holds. At very high temperatures, deviations from Eq. (34) become evident [see, maroon up-triangles in (**b**) and (**c**) for the case $T = 2.0$]. Source data are provided as a Source Data file.

where $D_0$ and $A$ are constants. The parameter $A$ in Eq. (35) is important since it determines how rapidly $D$ decreases with decreasing ($TS_{IS}$) [i.e., upon cooling]. Physically, the parameter $A$ has been associated with the fragility of the liquid. For example, in the case of the classical LJBM, both the parameter $A$ and the fragility of the LJBM increase with density[43]. Eq. (35) implies that the diffusivity at a given temperature is controlled by the corresponding number of IS available to the system in the PEL, i.e., the topography of the PEL controls the dynamics of the liquid of interest. Eq. (35) has been validated in computational studies of diverse classical liquids, including silica[32] and water[47]. Here, we show that Eq. (35) also holds for the LJBMs studied, independently of the quantum character ($h$) of the mixtures.

Figure 6 shows $D$ as a function of $1/TS_{IS}$ for the classical and $S_{IS}(T)$ for the classical and quantum LJBMs studied. For the classical LJBM, the values of $S_{IS}(T)$ are obtained from thermodynamic integration, following the procedure of refs. 43,53 [$S_{IS}(T)$ is shown in Fig. S5a of the SI]. For the quantum LJBMs, the values of $S_{IS}(T)$ in Fig. 6 are taken from the corresponding expression in the PEL formalism, Eq. (10) and Fig. 4a, and using the equilibrium IS energy [$e_{IS} \rightarrow E_{IS}$] given by Eq. (28). The resulting expressions for $S_{IS}(T)$ are evaluated at the same temperatures at which the diffusion coefficient $D(T)$ is calculated from the MD/PI computer simulations. The diffusion coefficient is calculated from the mean-square displacement (MSD) of the ring-polymer centroids as a

function of time. Specifically, for an atomistic liquid, MSD($t$) ≈ $6Dt$ at long-times[56]. Also included in Fig. 6 are the linear fits to the data points using Eq. (35) (solid lines). The agreement between the RPMD simulations and the Adam-Gibbs equation is remarkably good, extending for approximately four orders of magnitude in $D$. Interestingly, the activation parameter $A$ in the AG relation [Eq. (35)] increases with increasing $h$. This suggests that the fragility of the LJBMs studied also increases as the system becomes more quantum.

We note that there are a few theoretical expressions for liquids, including modifications of the AG relationship, that relate $D$ and $S_{IS}$ (see, e.g., refs. 57–59). It would be interesting in the future to study how these expressions compare with the AG relationship in the case of liquids that obey quantum mechanics, particularly at very low temperatures close to the glass transition[60–62].

## Discussion

One of the main goals of this work is to show that, from a thermodynamic/statistical mechanics point of view, the PEL formalism provides a unifying description of classical and quantum liquids over a wide range of temperatures down to the glass state. In the classical case, the PEL of a liquid is given by the potential energy function of the system as a function of the atoms coordinates, $U(\mathbf{r}_1, \mathbf{r}_2, ..., \mathbf{r}_N)$, and the system is represented by a point moving on such a PEL (CL-PEL). In the

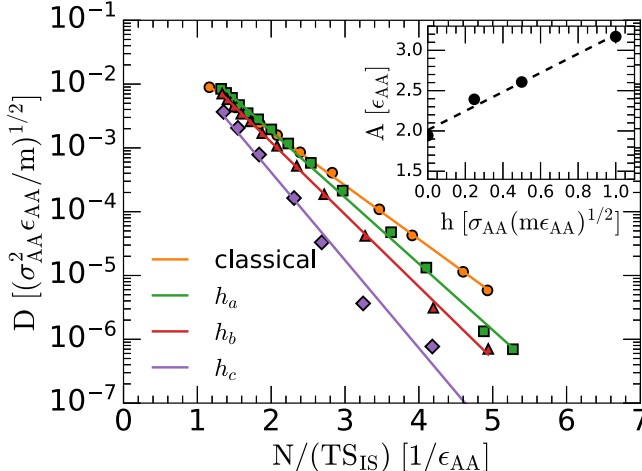

**Fig. 6 | Adam-Gibbs relation for the classical and quantum LJBMs.** Diffusion coefficient of the classical and quantum LJBMs with $h = h_a$, $h_b$, $h_c$. Symbols are the results from the MD/RPMD simulations with the values of $S_{IS}(T)$ given by the lines in Fig. 4b at the corresponding temperatures (see text). The solid lines are the linear fits based on the Adam-Gibbs relation, Eq. (35). The Adam-Gibbs relation holds remarkably well for all the classical and quantum LJBMs studied for over more than four decades in $D$. Inset: Parameter $A$ defined in the AG equation [Eq. (35)] for the LJBMs studied using different values of Planck's constant $h$; the value of $A$ obtained from classical MD simulations is indicated by the circle at $h = 0$. $A(h)$ increases approximately linearly with increasing $h$ ($h \leq 1$), as the system becomes more quantum. Source data are provided as a Source Data file.

quantum case, one can associate the PEL of the underlying ring-polymer system (RP-PEL) to the quantum liquid (for a fixed number of beads per ring-polymer, $n_b$). As discussed in this work, the PEL associated to the quantum liquid is defined by Eq. (5), and the quantum liquid, again, can be represented by a point moving on the so-defined RP-PEL. Therefore, independently of whether the system obeys classical or quantum mechanics, the liquid is characterized by a point that describes a trajectory on the corresponding CL-PEL/RP-PEL as the system visits different basins over time. In the glass state, the representative point of the system would be limited to vibrational motion within a single basin of the CL-PEL/RP-PEL as molecules/atoms are unable to diffuse.

Building upon the similarities between classical and quantum liquids in the PEL formalism, we show that the RP-PEL can be used to define a configurational entropy that can be associated to the quantum liquid in a precise manner. As for the classical case, $S_{IS}(N, V, T)$ is a measure of the number of IS available to the ring-polymer system associated to the quantum liquid (Eq. (10)). Importantly, using the so-defined configurational entropy, it is shown that one can also extend the Adam-Gibbs relation to the quantum case. This implies that the dynamics of the quantum liquid is controlled by the number of IS available to the ring-polymer system at a given $(N, V, T)$ [Fig. 6]. It follows that the configurational entropy associated to the quantum liquid has a physical relevance from the dynamical point of view.

The configurational entropy defined in this work is based on the hypothesis (B), that the distribution of IS, $\Omega_{IS}$, at a given working conditions $(N, V, T)$, is identical for the classical and quantum liquid. Within the Gaussian approximation of the PEL (Eq. (9)), which is valid for the studied LJBMs (Fig. 3), this also implies that the PEL quantities $\{\alpha, E_0, \sigma^2\}$ do not depend on the nature (quantum/classical) of the liquid studied, and that they depend only on $V$ (but not on $T$). Accordingly, the CL-PEL and RP-PEL have the same number of IS at a given IS energy/PEL depth. Hence, from a practical point of view, the PEL quantities $\{\alpha, E_0, \sigma^2\}$ can be calculated from classical MD

simulations (as opposed to the more computationally expensive path-integral computer simulations). We stress that the $S_{IS}(N, V, T)$ for the quantum LJBMs studied in this work are validated (a) numerically, via thermodynamic integration and RPMD simulations [Fig. 4b], and (b) by testing that Eq. (34) is in agreement with the RPMD simulations of the LJBMs. It follows that hypothesis (B) is self-consistent within the PEL formalism. As explained in this work, hypothesis (B) should hold as far as the ring-polymers associated to the quantum liquid collapse at the IS. This seems to be rather general for systems where the NQE (atoms delocalization) are mild; indeed, PIMD simulations of liquid and glassy water[13,63], monatomic quantum liquids[41,42,64], and LJBM (this work) at low temperatures show no sign of delocalized ring-polymers at the corresponding IS.

This study emphasizes the role of anharmonic contributions in the PEL formalism. Our RPMD simulations show that while the PEL of the classical and weakly-quantum LJBMs are harmonic ($h \leq h_a$), anharmonicities become increasingly relevant with increasing $h$ ($h > h_a$); see Fig. 2. Accordingly, in this study, we also provide a general approach to include anharmonic contributions in the PEL formalism. In this regard, we note that Eqs. (19) and (21) define unequivocally the anharmonic Helmholtz free energy since these are two independent equations that define the two partial derivatives of $F_{vib}^{anh}(N, V, T, e_{IS})$ (at fixed $(N, V)$). We note that, in the case of classical liquids, recent studies found that anharmonic corrections to the configurational entropy are crucial to correctly describe the glassy dynamics of the LJBM[61,62]. Our results for the quantum LJBMs with $h = h_b$, $h_c$ are consistent with these studies.

Being able to extend the PEL formalism to liquids/glasses that obey quantum mechanics provides a useful tool to explore the role of quantum mechanics on atomistic liquids composed of light-element as well as molecular liquids composed of small molecules that contain H atoms, such as water. In these cases, NQE can play a relevant role even at non-negligible temperatures (e.g., $T \approx 80 - 270$ K)[65]. In particular, extending the concept of configurational entropy to quantum liquids is fundamental in providing a thermodynamic description of liquids that obey quantum mechanics. For example, it is possible to write an equation of state for quantum liquids using the PEL formalism by following a procedure similar to that used in the case of classical liquids[36,39]. We note that the PEL formalism for quantum liquids allows one to characterize quantitatively the changes in the thermodynamic and dynamical properties of a liquid as the quantumness of the system (as quantified by the Planck's constant $h$) is varied. However, the PEL formalism does not provide information on the physical mechanisms underlying such variation. For example, it remains unclear what physical mechanisms in the LJBMs studied result in the suppression of the atoms diffusivity and the increase of $T_K$ when $h$ increases. Addressing these issues is important and warrants further investigation.

## Methods

In this work, we perform classical MD and RPMD simulations of the LJBM defined in ref. 66. The LJBM model liquid is a good glass-former that has been studied extensively in the past using classical MD simulations[43,67,68]. We consider a LJBM composed of $N_A = 800$ type-A and $N_B = 200$ type-B atoms interacting via Lennard-Jones pair interaction potentials $V_{\alpha\beta}(r) = 4\epsilon_{\alpha\beta}[(\sigma_{\alpha\beta}/r)^{12} - (\sigma_{\alpha\beta}/r)^6]$ with $\alpha\beta \in \{A, B\}$. Following ref. 66, we use reduced units where $\sigma_{AA} = 1.0$, $\sigma_{BB} = 0.80$, $\sigma_{AB} = 0.88$ (length), $\epsilon_{AA} = 1.0$, $\epsilon_{BB} = 0.5$, $\epsilon_{AB} = 1.5$ (energy), and $m_A = m_B = m = 1.0$ (mass); LJ pair interactions are truncated and shifted at a cutoff distance $r_{c,\alpha\beta} = 2.5\,\sigma_{\alpha\beta}$, as implemented in the original work of Kob and Andersen[66]. Moreover, we also add a switching function to the pair interaction potentials, as implemented in OpenMM[69], to make the forces smooth functions of the inter-particle distance $r$. The switching function only affects the LJ pair interaction potential at inter-particle distances $r_{s,\alpha\beta} < r < r_{c,\alpha\beta}$ where $r_{s,\alpha\beta} = 0.9\,r_{c,\alpha\beta}$. The system box is cubic and periodic boundary conditions apply in all three directions.

As for the case of path-integral molecular dynamics (PIMD) simulations, the RPMD technique is based on the path integral formulation of quantum statistical mechanics, and both techniques can be used to calculate thermodynamic (e.g., pressure) and structural properties (e.g., radial distribution functions) of a liquid in the presence of NQE. In PIMD/RPMD simulations, each atom of the system is represented by a ring-polymer composed of $n_b$ beads; by setting $n_b = 1$, the PIMD/RPMD simulation technique reduces to classical MD simulations. To calculate (approximately) the diffusion coefficient, $D(T)$, we use the RPMD technique and (constant-temperature) RPMD simulations. As explained in refs. [70,71], from the RPMD trajectory, one can calculate, approximately, the dynamical properties of the quantum system. In this work, $D(T)$ is calculated from the long-term behavior of the mean-square displacement of the ring-polymer centroids obtained from the RPMD simulations at constant temperature (see, e.g.,[13,54,72]).

To understand the differences in the PEL properties of the quantum and classical LJBM, we follow refs. [41,42] and study a family of LJBMs each characterized by a different value of the Planck's constant $h$. The quantum character of a liquid increases with increasing values of $h$ since the atom delocalization becomes more pronounced as $h$ increases (see Fig. 1d). In this study, we consider the cases $h_a = 0.2474$, $h_b = 0.5000$, and $h_c = 1.0000$ (in reduced units of $\sigma_{AA}(\epsilon_{AA}m)^{1/2}$). We note that the values of $h = h_a, h_b, h_c$ considered here are not negligible; for example, as discussed in detail in ref. [42], one obtains $h \approx 1.78$ and $h \approx 0.18$ (in reduced units) for the case of $H_2$ and argon, respectively [see also Table S1 of the Supplementary Information (SI)]. For reference, we also include the results for the classical LJBM.

Most of the RPMD simulations are performed with $n_b = 10$ beads per ring-polymer. However, to test for $n_b$-effects, we also run RPMD simulations using $n_b = 20, 40$ ($h = h_c$). Classical MD simulations are run by setting $n_b = 1$ in the RPMD simulations. All computer simulations are performed at constant $(N, V, T)$ for the density $\rho = 1.2$ ($V/N = 9.4$, $N = 1000$) and extend over a wide range of temperatures. Additional computer simulations at constant $(N, V, T)$ are performed at $T = 5.0$ and over a wide range of volumes to calculate the total entropy of the system (Sec. III of the SI). The pressure is evaluated using the estimator given in Eq. (12.8.43) of ref. [73]. The temperature is controlled by using a local PILE thermostat with stochastic collision frequency $\gamma = 0.6$. At a given state point, the starting configuration of the RPMD simulation is taken from an independent equilibrium simulation performed at a very high temperature, $T = 5.0$. For most state points studied, equilibration runs extend for $t_{eq} \geq 100\,\tau_\alpha$ where $\tau_\alpha$ is the relaxation time of the system defined as the time at which the mean-square displacement (MSD) of the type-A atoms is $1.0\,\sigma_{AA}^2$. To obtain a large number of IS, we perform very long production runs, of $t \geq 1000\tau_\alpha$ (see below). At the lowest temperatures ($T = 0.55, 0.60$ for $h_c$; $T = 0.435$ for $h_a$ and classical MD), total equilibration and production runs last for 20 ~ 500$\tau_\alpha$ due to the extremely slow dynamics of the system. All the RPMD simulations are performed using the OpenMM (version 7.3.0) software package[69] modified to include the target Planck's constant $h$. In the OpenMM computer simulations, the atoms mass is set to $m_A = 39.948$ amu, $\sigma_{AA} = 1.0$ Å, and $\epsilon_{AA} = 1.0$ kJ/mol (but all quantities are reported in reduced units). The simulation time step for the classical MD and RPMD simulations for $h = h_a, h_b$ is $dt = 0.01$ ps ($T \leq 1.0$) and $dt = 0.001$ ps ($T \in (1.0, 5.0]$); in the case of the RPMD simulations for $h = h_c$, $dt = 0.005$ ps ($T \leq 1.0$) and $dt = 0.0005$ ps ($T \in (1.0, 5.0]$).

From the long RPMD production runs, at a given temperature, we extract more than 1000 independent configurations. Two configurations are considered to be independent if they are separated in time by, at least, $\delta t = \tau_\alpha$. For each of these configurations, we calculate the corresponding IS by minimizing the potential energy of the system using the L-BFGS-B algorithm (the potential energy minimization is performed using the *scipy.optimize* routine in Python[74] with an error tolerance of $10^{-12}$, with the forces and total potential energy obtained

from the OpenMM software package using the OpenCL platform with double precision). The IS energy is obtained directly from the minimization algorithm. For each IS, we also calculate analytically the components of the mass-weighted Hessian matrix. The mass-weighted Hessian matrix is then diagonalized to calculate the corresponding eigenvalues which provide the normal mode vibrational frequencies of the system (diagonalization is performed using the *Numpy* linear algebra package in Python[75]).

## Data availability
The data supporting the findings of this study are available within the article and its Supplementary Information. Source data are provided with this paper.

## Code availability
The molecular dynamics and path-integral simulations were performed using the OpenMM software package[69], which is publicly available at (https://openmm.org) The potential energy minimizations and normal mode analyses were performed using standard Python libraries (SciPy[74] and NumPy[75]).

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

## Acknowledgements

This work was supported by the NSF CREST Center for Interface Design and Engineered Assembly of Low Dimensional systems (IDEALS), NSF grant number HRD-1547380. A. E. is supported by the NSF CREST Postdoctoral Research Program under Award No. 2329339. This work was also supported by the Advanced Cyberinfrastructure Coordination Ecosystem: Services & Support (ACCESS) program[76] which is supported by the National Science Foundation Grant Nos. 2138259, 2138286, 2138307, 2137603, and 2138296. Y.Z. is thankful for financial support from The Grace Spruch '47 Fund at the Physics Department of Brooklyn College.

## Author contributions

Y.Z., A.E., G.E.L., and N.G. contributed equally to the design of the study, the analysis and interpretation of results, and the writing of the manuscript.

## Competing interests

The authors declare no competing interests.
