## [Transparent Peer Review file · Nature Communications]

Configurational Entropy and Adam-Gibbs Relation for Quantum Liquids

Corresponding Author: Mr Yang Zhou

Version 0:

Reviewer comments:

Reviewer #1

(Remarks to the Author)

Review of "Configurational Entropy and Adam-Gibbs Relation for Quantum Liquids" by Yang Zhou, Ali Eltareb, Gustavo E. Lopez, Nicolas Giovambattista NCOMMS-25-32290-T

Zhou et al. have presented an interesting analysis of the Adam-Gibbs (AG) relation between dynamics and thermodynamics in a model glass-forming liquid. They have extended a potential energy landscape (PEL) based formalism that enables computing configurational entropy S_c for a quantum liquid obeying Boltzmann statistics, while doing classical simulation. This is based on a formally exact mapping from a classical to a quantum many-body Hamiltonian system that may be traced to Chandler and Wolynes, JCP, 74, 4078 (1981). They also characterize dynamics at low temperature using diffusion coefficient D . Thus the authors are able to test the validity of the AG relation in presence of explicit quantum effects.

One of the outstanding challenges in the field of glass-forming liquids is to close the gap between the capabilities of experiments and computer simulations. The formalism applied in the present manuscript (MS) offers one approach to make computer simulations more realistic by explicitly considering quantum nature of the atomic motion. However, I have some queries regarding the method and the conclusions, as detailed below, which the authors are requested to clarify.

- 1) Although the LJBM system studied here is a prototype (classical) computer glass and hence is certainly important to characterize, usually their experimental counterparts are colloids and metallic glasses, i.e. typically consisting of heavier elements than in hydrogen, helium and water. It would be interesting to know to what extent the "nuclear quantum effects" are significant for the LJBM model.
- 2) One of the challenges for classical simulations is to sample lower temperatures or longer timescales in equilibrium, see e.g. Berthier and Reichmann, Nat. Rev. Phys., 5, 102 (2023). In the PEL language this is equivalent to accessing lower energy basins. Figs 2a of the MS shows that increasing Planck's constant h lowers the inherent structure energy. From a purely computational point of view, does the present formalism offer a way to efficiently sample lower temperatures / longer timescales in equilibrium?
- 3) A significant portion of the formalism to compute configuration entropy has already been presented by the authors in detail in Refs 13, 41, 42. For example, configuration entropy in harmonic approximation has been discussed in Ref. 41. Even the observation that the anharmonic correction is essential to incorporate was discussed in Ref. 42, although I find the discussion on anharmonic correction to be more complete in the present MS. Nevertheless, I feel that the description of the PEL formalism may be shortened and only the part that is new should be elaborated.
- 4) The lowest temperature T studied in the present MS seems to be 0.435 in reduced LJ units. This is a relatively higher temperature. There are recent classical computational studies which finds deviation from the AG relation at lower temperatures e.g. Ortlieb et al. N. Comm., 14, 2621 (2023) which however does not include anharmonic correction. Other groups have reported that the AG relation remains valid even at lower T provided that the anharmonic corrections are included, see Das and Sastry, J. Non-Cryst. Solids: X, 14, 100098 (2022). See also Ozawa et al. J. Chem. Phys., 151, 084504 (2019). Given this background, the validity of the AG relation in the LJBM system seems to be an open question. In particular, looking at the fits and the data provided in Fig. 7, could it be possible that there is a systematic deviation from the AG relation at the lowest temperature for different h ?

It would be also useful to check whether the more general random first order transition theory [RFOT, J.-P. Bouchaud and G. Biroli, J. Chem. Phys. 121, 7347 (2004); V. Lubchenko and P. G. Wolynes, Annu. Rev. Phys. Chem. 58, 235 (2007); T. R. Kirkpatrick and D. Thirumalai, Rev. Mod. Phys. 87, 183 (2015)] works better than the AG relation for the quantum liquids with physically meaningful exponent values.

5) The present formalism utilizes the statistical properties of the PEL minima but not that of the barrier height distribution which requires computation of the saddle points. However, according to the AG relation, knowledge of the activation energy parameter A is also required to determine the diffusion coefficient D . This point may be clarified in the introduction / conclusion sections. Fig. 7 suggests that A depends on h . Since Sc is assumed to be independent of h , does it imply that D is a function of T/h ?

6) A question that naturally arises is to what extent the glassy dynamics at low temperatures is affected by quantum effects such as tunneling between two basins. E.g. Ref. 14 has presented evidence that the glass transition temperature T_g is different for H₂O and D₂O and has attributed it to tunneling. The present formalism is well placed to quantify the effects of “quantumness” through dependence on h and perhaps the number of ring-polymer beads n_b . However, the authors are requested to clarify whether tunneling between two basins is physically relevant for the LJBM system studied here.

In particular, a naive expectation is that such tunnelling will provide an additional mechanism for structural relaxation and hence should speed-up dynamics at the same temperature. However, Fig. 7 of the present MS shows that the diffusion coefficient D decreases with increase in the Planck’s constant h at the same temperature T . Also, the activation energy A in the AG relation increases with increasing h . These trends imply a slower dynamics for larger h , which is not clear to me.

7) In addition to h , n_b seems to be a second independent parameter to tune the degree of “quantumness” of the system. It would be desirable to see the n_b dependence of D to understand how the quantum nature affects dynamics.

8) I would also like to request the authors to provide the details of how MSD varies with h , T/h and n_b and whether taking into account only species A (or B) particles vs. all the particles make any difference in the conclusions.

9) It will be useful to see how the characteristic temperatures e.g. the onset temperature T_{onset} , the VFT divergence temperature T_0 and the Kauzmann temperature T_k depends on the degree of quantumness, h and n_b . This will enable direct comparison with experiments such as Ref. 14. It will also offer new insights to the ongoing debates about whether the ideal glass transition should occur at zero or finite temperatures [see e.g. Berthier et al., Nat. Commun. 10, 1508 (2019); Nath and Sengupta, J. Chem. Phys. 161, 034504 (2024)]. In Fig. 5, the authors have noted that T_k increases with increasing h . What could be the physical explanation for this trend?

10) At a conceptual level, the physical meaning of tuning the Planck’s constant h is not very clear to me. The spring constants in the second term of the Hamiltonian in Eqn. 4 is a function of T/h . Does it imply that T/h should be the control parameter instead of T and h independently? Specifically, is there any data collapse of E_{IS} , E_{vib} in Figs 2 or 4, Sc in Fig 5 and D in Fig 7 plotted against T/h ? And more generally, by some combination of T , h and n_b ?

11) One of the key assumptions in the present analysis is that the distribution of IS minima with energy e_{IS} should be independent of h . The authors have argued that this should be the case based on the numerical observation of the collapse of ring-polymers in the IS, [Refs 41, 42]. However, this implies that the parameters $\{\alpha, E_0, \sigma\}$ should be independent of h as well. This is however, in contradiction to the evidence shown in earlier works, e.g. inset of Fig. 4a in Ref. 41. Perhaps the authors consider sharpening the explanations to rationalize the choice of option B in sec 3.3.

In particular, consider what happens when the quantum liquid is gradually cooled towards T_g . Since the ring-polymer Hamiltonian in Eqn. 4 is explicitly T dependent, the corresponding ring-polymer PEL should continuously deform as T is decreased. As a result, in principle, it is possible that some of the energy barriers goes to zero at some particular T and thus an IS minimum disappears. This would then change the distribution of IS minima. How such a possibility is excluded, is not clear to me.

12) Even for purely classical computations, the Planck’s constant h do appear in the total entropy calculation through the ideal gas contribution. Thus for computational purpose h is non-zero even for classical systems e.g. $h \sim 0.18$ in Argon units. There is a possibility that the total entropy may become negative at finite temperatures for increasing h , violating the third law of thermodynamics. Hence I would like to request for total entropy data for different h (or T/h). This may easily be computed by standard techniques such as thermodynamic integration, see e.g. Nandi et al. J. Chem. Phys. 156, 014503 (2022) for a critical assessment of the method.

13) Is the $Sc(T)$ in Fig. 7 averaged over IS energies sampled at a given temperature? This may be clarified.

14) Using $\ln Q$ instead of $c(T)$ in Eq. 34 and using different symbols to represent different h in Figs. 1, 2, 4, 7 may improve readability.

15) For the sake of completeness, the formula to compute pressure P in Fig. 1 should be defined.

Reviewer #2

(Remarks to the Author)

The manuscript presents a rigorous extension of the potential energy landscape (PEL) formalism developed for classical liquids to quantum liquids, where at the quantum level, each particle is represented by a ring polymer. The system studied is the well known Lennard-Jones binary mixture, the Kob Andersen model. This work is built upon their prior foundational work, Refs 13,41,42, where they studied the PEL of quantum liquids. In the present study, they have extended the formalism to study the configurational entropy and investigate the validity of the Adam-Gibbs relationship between diffusion and configurational entropy. The important observations are the following. The configuration entropy expression of the classical and quantum liquids remains the same because, in the inherent state, the ring polymer collapses to a point (shown in the earlier studies for different systems). However, for the quantum system, the inherent structure energy at a particular temperature is dependent on Planck's constant 'h'. They also show that for high values of 'h', the anharmonic contributions of the PEL become important. Since the inherent structure energy is different for different values of 'h', the temperature dependence of the configurational entropy is different for different quantum systems. As the quantumness 'h' is increased, the configurational entropy disappears at a higher temperature. They have also shown that for all the systems, the Adam-Gibbs relationship holds spanning data over multiple orders of magnitude. This is an important observation as it clearly shows that the dynamics of a quantum liquid depends on the available inherent energy states at that temperature. The use of ring polymer molecular dynamics (RPMD) combined with PEL theory for quantum liquids is still relatively unexplored, giving the work a high degree of originality. The variation of the configurational entropy and the Adam-Gibbs relationship with h are interesting results and are shown for the first time. This study opens a new direction where the theories of glassy transition can now be applied to quantum systems.

Thus, I would like to state that this work is highly original and contributes a new direction for applying glass transition theories to quantum systems, a domain that has been largely underexplored. The methodology is rigorous, and the simulation parameters and analytical assumptions are clearly stated. The theoretical framework is well constructed, and the numerical consistency checks (e.g., Eq. 33 validation) add credibility to the conclusions.

I fully support the acceptance of this paper in Nature Communications. However, I would like the Authors to have some discussion on the following points.

Why does the configurational entropy decrease with h? Numerically, we know that the effect comes from the inherent structure energy, but a few lines of discussion will be nice to have. Also, the Authors mention that "Our RPMD simulations show that SIS shifts towards higher temperatures as h increases." I think it will be better to state that "Our RPMD simulations show that at a particular temperature, the SIS decreases with increasing h". I would also like the Authors to discuss why the activation energy "A" in the Adam-Gibbs relationship increases with h and what it physically signifies.

Version 1:

Reviewer comments:

Reviewer #1

(Remarks to the Author)

Review of the revised manuscript "Configurational Entropy and Adam-Gibbs Relation for Quantum Liquids" [NCOMMS-25-32290A] by Yang Zhou, Ali Eltareb, Gustavo E. Lopez, Nicolas Giovambattista

I have gone through the revised manuscript and the referee replies by Zhou et al. The revised version has a clearer exposition, additional details about the analysis and the results are now more convincing. Hence, I recommend its acceptance in Nature Communications. A few minor points are mentioned below:

1) In Fig. 1d, the legend for "classical" ($h=0$) should be put above h_a , similar to e.g. Fig 3a.

2) Typo in line 690: $S_{IS}(T)$ in are  $S_{IS}(T)$ are

3) Last but not the least, several groups using diverse approaches have analysed the effect of anharmonic corrections to the potential energy landscape (PEL) and the configurational entropy, especially in the last few years and for the LJBM model. They have come to the same conclusion that the anharmonic correction to the configurational entropy is crucial to correctly describe glassy dynamics of classical liquids, in particular of the LJBM model. While it is reassuring that the authors find that it is also true for quantum liquids, the glass physics community sometimes are not aware of the importance of including the anharmonic corrections. Thus, these literatures should be acknowledged.

Reviewer #2

(Remarks to the Author)

As I mentioned earlier, the use of ring polymer molecular dynamics (RPMD) combined with PEL theory for Quantum liquids remain relatively unexplored, lending the work a high degree of originality.

The authors have answered all the questions raised by the first and second reviewers. However, in the manuscript and also in their replies to the questions, I feel that although what happens when quantumness is varied is well-documented, the physical understanding of the variation in quantumness remains unclear. Perhaps the authors could include a discussion in this direction.

Reviewer #1 (Remarks to the Author):

Review of “Configurational Entropy and Adam-Gibbs Relation for Quantum Liquids” by Yang Zhou, Ali Eltareb, Gustavo E. Lopez, Nicolas Giovambattista NCOMMS-25-32290-T.

Zhou et al. have presented an interesting analysis of the Adam-Gibbs (AG) relation between dynamics and thermodynamics in a model glass-forming liquid. They have extended a potential energy landscape (PEL) based formalism that enables computing configurational entropy S_c for a quantum liquid obeying Boltzmann statistics, while doing classical simulation. This is based on a formally exact mapping from a classical to a quantum many-body Hamiltonian system that may be traced to Chandler and Wolynes, JCP, 74, 4078 (1981). They also characterize dynamics at low temperature using diffusion coefficient D . Thus the authors are able to test the validity of the AG relation in presence of explicit quantum effects.

One of the outstanding challenges in the field of glass-forming liquids is to close the gap between the capabilities of experiments and computer simulations. The formalism applied in the present manuscript (MS) offers one approach to make computer simulations more realistic by explicitly considering quantum nature of the atomic motion. However, I have some queries regarding the method and the conclusions, as detailed below, which the authors are requested to clarify.

We thank the reviewer for reading the manuscript in detail and for raising the insightful points listed below. We are glad that the reviewer finds the manuscript interesting and values the approach implemented in our work to study glass-forming liquids in the presence of quantum effects.

1) Although the LJBM system studied here is a prototype (classical) computer glass and hence is certainly important to characterize, usually their experimental counterparts are colloids and metallic glasses, i.e. typically consisting of heavier elements than in hydrogen, helium and water. It would be interesting to know to what extent the “nuclear quantum effects” are significant for the LJBM model.

The reviewer is raising an important point. At present, it is not clear whether the LJBM *per se* is relevant as a model system to study realistic quantum liquids. But it is possible that binary mixtures of isotopes or light elements, such as H₂ and He, could exhibit some of the properties observed in the quantum LJBM systems studied (the LJ pair potential interactions may have to be reparametrized).

Our motivation to study the LJBM is that, as the reviewer states, this is a prototypical well-studied, classical glass former that serves as an ideal model system to explore the role of quantum fluctuations (nuclear quantum effects, NQE). Our choice is also motivated by the work of Markland et al. [Markland, T. E. et al. Quantum fluctuations can promote or inhibit glass formation. Nature Physics 7, 134–137 (2011)] as well as Biroli and Zamponi [Biroli, G., Zamponi, F. A Tentative Replica Theory of Glassy Helium 4. J Low Temp Phys 168, 101–116 (2012)] who investigated the role of quantum fluctuations on the dynamics of hard spheres, LJBM, and LJ systems.

2) One of the challenges for classical simulations is to sample lower temperatures or longer timescales in equilibrium, see e.g. Berthier and Reichmann, Nat. Rev. Phys., 5, 102 (2023). In the PEL language this is equivalent to accessing lower energy basins. Figs 2a of the MS shows that increasing Planck's constant h lowers the inherent structure energy. From a purely computational point of view, does the present formalism offer a way to efficiently sample lower temperatures / longer timescales in equilibrium?

We thank the reviewer for raising this important question. At present, it is unclear whether path-integral computer simulations may allow one to explore more stable configurations, with lower IS energy, than classical MD simulations. While Fig 2a suggest that, at a given temperature, the quantum liquids can access deeper regions of the PEL than their classical counterpart, the diffusion coefficient of the LJMB also decreases with increasing h (for the range of h -values studied here); see also Ref. [Markland, T. E. et al. Quantum fluctuations can promote or inhibit glass formation. Nature Physics 7, 134–137 (2011)]. We agree that the point raised by the reviewer is important, and we will address this issue in detail in a future study.

On a side note, Fig. 2a indicates that the range of IS energies sampled by the LJBM is (-7.70, -7.55), independently of h . Therefore, at least based on our results, adding quantum fluctuations does *not* allow the LJBM to sample deeper regions in the PEL than those already accessible to the classical liquid.

3) A significant portion of the formalism to compute configuration entropy has already been presented by the authors in detail in Refs 13, 41, 42. For example, configuration entropy in harmonic approximation has been discussed in Ref. 41. Even the observation that the anharmonic correction is essential to incorporate was discussed in Ref. 42, although I find the discussion on anharmonic correction to be more complete in the present MS. Nevertheless, I feel that the description of the PEL formalism may be shortened and only the part that is new should be elaborated.

We appreciate the reviewer's suggestion and, indeed, we attempted to shorten the theory section on the PEL formalism. However, we also found that by doing so, the discussion on (i) the calculation of the configurational entropy and (ii) the treatment of the "anharmonic contributions" becomes obscure and difficult to follow. Our manuscript covers, for the first time, the treatment of both (i) and (ii) in the PEL formalism for quantum liquids [Refs. 13, 41, 42 do not do so]. In particular, the realization that the classical and quantum liquid should have the same configurational entropy, $S_{IS}(e_{IS})$, is first stated in the submitted manuscript (and absent in Refs. 13, 41, 42). The treatment of anharmonic contributions is usually discussed lightly in PEL studies of *classical* liquids and our presentation on this topic differs from previous studies.

4) The lowest temperature T studied in the present MS seems to be 0.435 in reduced LJ units. This is a relatively higher temperature. There are recent classical computational studies which finds deviation from the AG relation at lower temperatures e.g. Ortlieb et al. N. Comm., 14, 2621 (2023) which however does not include anharmonic correction. Other groups have reported that the AG relation remains valid even at lower T provided that the anharmonic corrections are included, see Das and Sastry, J. Non-Cryst. Solids: X, 14, 100098 (2022). See also Ozawa et al. J. Chem. Phys., 151, 084504 (2019). Given this background, the validity of the AG relation in the LJBM system seems to be an open question. In particular, looking at the fits and the data provided in Fig. 7, could it be possible that there is a systematic deviation from the AG relation at the lowest temperature for different h ?

It would be also useful to check whether the more general random first order transition theory [RFOT, J.-P. Bouchaud and G. Biroli, J. Chem. Phys. 121, 7347 (2004); V. Lubchenko and P. G. Wolynes, Annu. Rev. Phys. Chem. 58, 235 (2007); T. R. Kirkpatrick and D. Thirumalai, Rev. Mod. Phys. 87, 183 (2015)] works better than the AG relation for the quantum liquids with physically meaningful exponent values.

We thank the reviewer for these useful suggestions. We agree that it is possible that the AG relation for quantum LJBM may not hold at lower temperatures than those studied, and we are not ruling out this possibility. Nevertheless, Fig. 6 (Fig. 7 in the previous version of the manuscript) shows that AG relation holds excellently over at least 4 decades in the diffusion coefficient for the quantum LJBM studied with different \hbar , which is remarkable. Unfortunately, PIMD simulations are much more computationally expensive than classical MD simulations. In addition, exploring lower temperatures for quantum LJBM than those explored in our study requires a larger number of beads per ring-polymer as well as increasingly longer simulation times. The PI computer simulations at $\hbar = \hbar c$ and lowest 2 temperatures may deviate from the AG but we cannot discard the need for longer simulation times and/or larger number of beads for these two state points. While the AG relationship may breakdown at lower temperatures, or may be less precise than other theoretical predictions, it is nonetheless remarkable how well it compares with our path-integral computer simulations (Fig. 6). A test of our results with predictions other than the AG relationship is important but it falls out of the scope of our study. We prefer to keep the manuscript focused, showing that the techniques and concepts based on the PEL formalism learnt by the liquids/glasses community over 40-50 years, using *classical* systems, can be extended to the case of quantum liquids/glasses with relatively small conceptual modifications.

At the end of Sec. 4 of the new version of the manuscript, we add a new paragraph addressing the reviewer's point: *"We note that there are a few theoretical expressions for liquids, including modifications of the AG relationship, that relate D and S_{IS} (see, e.g., Refs [J.-P. Bouchaud and G. Biroli, J. Chem. Phys. 121, 7347 (2004); V. Lubchenko and P. G. Wolynes, Annu. Rev. Phys. Chem. 58, 235 (2007); T. R. Kirkpatrick and D. Thirumalai, Rev. Mod. Phys. 87, 183 (2015)]. It would be interesting in the future to study how these expressions compare with the AG relationship in the case of liquids that obey quantum mechanics, particularly at very temperatures close to the glass transition [Ortlieb et al. N. Comm., 14, 2621 (2023), Das and Sastry, J. Non-Cryst. Solids: X, 14, 100098 (2022), Ozawa et al. J. Chem. Phys., 151, 084504 (2019)]."*

5) The present formalism utilizes the statistical properties of the PEL minima but not that of the barrier height distribution which requires computation of the saddle points. However, according to the AG relation, knowledge of the activation energy parameter A is also required to determine the diffusion coefficient D . This point may be clarified in the introduction / conclusion sections. Fig. 7 suggests that A depends on \hbar . Since S_c is assumed to be independent of \hbar , does it imply that D is a function of T/\hbar ?

We agree with the reviewer that the activation parameter A in the AG relation is important (and depends on \hbar). In line below Eq. (35), we state that *"...where D_0 and A are constants. The parameter A in Eq. 35 is important since it determines how rapidly D decreases with decreasing $(T S_{IS})$ [i.e., upon cooling]. Physically, the parameter A has been associated with the fragility of the liquid. For example, in the case of the classical LJBM, both the parameter A and the fragility of the LJBM increase with density [Sastry, S. The relationship between fragility, configurational entropy and the potential energy landscape of glass-forming liquids. Nature 409, 164–167 (2001)]. Eq. 35 implies..."*

Motivated by the reviewer's comment, we add an inset to Fig. 6 (Fig. 7 in the previous version of the manuscript) showing $A(h)$ [the inset of Fig. 6 is reproduced in Fig. A below], and include the following text at the end of the second-to-last paragraph of Sec.4: "Interestingly, the activation parameter A in the AG relation [Eq. 35] increases with increasing h . This suggests that the fragility of the LJBM s studied also increases as the system becomes more quantum."

Fig. A. Parameter A defined in the AG equation for the LJBM s studied using different values of Planck's constant h ; the value of A at $h=0$ corresponds to the classical LJBM. $A(h)$ increases approximately linearly with increasing h ($h \leq 1$), as the system becomes more quantum.

We also test whether $D=D(T/h)$. As shown in the figure below, our results for different T and h do not collapse onto a single master curve, the reason being that, while $S_{IS}(e_{IS})$ is independent of h (Eq.10 of the main manuscript), the average IS energy $E_{IS}(T)$ does vary with h . Accordingly, $S_{IS}(T)=S_{IS}(E_{IS}(T))$ is a function of both T as well as (implicitly) h .

Fig. B. Diffusion coefficient of the LJBM s studied (A-type particles) for all temperatures and values of the Planck's constant h considered. D is plot as a function of T/h . Lines correspond to classical and quantum LJBM s with h_a , h_b , h_c (orange, green, red, and purple, respectively).

6) A question that naturally arises is to what extent the glassy dynamics at low temperatures is affected by quantum effects such as tunneling between two basins. E.g. Ref. 14 has presented evidence that the glass transition temperature T_g is different for

H2O and D2O and has attributed it to tunneling. The present formalism is well placed to quantify the effects of “quantumness” through dependence on \hbar and perhaps the number of ring-polymer beads n_b . However, the authors are requested to clarify whether tunneling between two basins is physically relevant for the LJBM system studied here.

The referee raises another interesting question. However, understanding tunneling using the PEL formalism is subtle.

For a single particle in 1D, moving along the x-axis, tunneling effects are usually explained in terms of the *classical* expression for the potential energy function of the particle, $V(x)$. Hence, interpreting tunneling in quantum liquids requires studying the quantum liquids using the PEL of the classical liquid counterpart (what we called classical PEL (CL-PEL) in Ref. [41]. In the CL-PEL, the system is represented by n_b points that travel over time (where n_b =number of beads per ring-polymers). Each of these n_b points in the CL-PEL are defined by the coordinates of one-and-only-one of the n_b replicas. In addition, these n_b representative points on the CL-PEL are connected by “springs” [41]. In our manuscript, we are using the PEL of the ring-polymer system associated to the quantum liquid of interest (briefly, the RP-PEL [41]). In the RP-PEL, the system is represented by a single point that travels over time. Hence, tunneling is difficult to “visualize” using the RP-PEL. We also note that, even if one uses the CL-PEL to study tunneling in liquids, it is not trivial how to disentangle the effects of (i) temperature from (ii) tunneling (both (i) and (ii) drive the n_b replicas to visit multiple basins in the CL-PEL at a given T).

We agree with the reviewer that for sufficient large “quantumness” (large \hbar) and low temperatures, the role of tunneling in deep supercooled liquids is not well understood and is an important question for future investigation. One may expect that tunnelling will provide an additional mechanism for structural relaxation at low temperatures, and hence, should speed-up the dynamics of the system. We note, however, that the dynamics in the LJBM at a given *temperature* slows down with increasing “quantumness” (for the range of \hbar values studied here) suggesting that tunneling contribution to the dynamics of the LJBM studied does not play a relevant role. The counter-intuitive slowing-down in the dynamics of the LJBM was first shown by Markland et al. [Markland, T. E. et al. Quantum fluctuations can promote or inhibit glass formation. *Nature Physics* 7, 134–137 (2011), Markland, T. E. et al. Theory and simulations of quantum glass forming liquids. *Journal of Chemical Physics* 136, 074511–074511 (2012)] and is consistent with the work of Biroli and Zamponi [Biroli, G. & Zamponi, F. A Tentative Replica Theory of Glassy Helium 4. *J Low Temp Phys* 168, 101–116 (2012)] who studied the role of nuclear fluctuations on the LJBM and LJ system. In the new version of the manuscript, we include in Fig. 1, panel (c), the T-dependence of the diffusion coefficient for the classical/quantum LJBM studied (A-type particles). This figure is reproduced below.

Fig. 1(c). Diffusion coefficient $D(T)$ of the classical and quantum LJBM with $h=ha$, hb , hc (A-type atoms) [top-to-bottom]. Consistent with Ref. [Markland, T. E. et al. Quantum fluctuations can promote or inhibit glass formation. Nature Physics 7, 134–137 (2011),], at a given low temperature, $D(T)$ decreases with increasing quantumness, as quantified by h , for the range of h -values studied.

In line 8 of Sec. 4, we also include the following sentence: “Consistent with Ref. [Markland, T. E. et al. Quantum fluctuations can promote or inhibit glass formation. Nature Physics 7, 134–137 (2011), Markland, T. E. et al. Theory and simulations of quantum glass forming liquids. Journal of Chemical Physics 136, 074511–074511 (2012), Biroli, G., Zamponi, F. A Tentative Replica Theory of Glassy Helium 4. J Low Temp Phys 168, 101–116 (2012)], at a given temperature, $D(T)$ decreases with increasing values of h , for the range of h -values studied, which is counterintuitive since NQE are expected to speed-up the dynamics. The NQE at low temperatures are...”

Reference

[41] Giovambattista, N. & Lopez, G. E. Potential energy landscape formalism for quantum liquids. Physical Review Research 2, 043441 (2020).

7) In addition to h , n_b seems to be a second independent parameter to tune the degree of “quantumness” of the system. It would be desirable to see the n_b dependence of D to understand how the quantum nature affects dynamics.

The number of beads per ring polymer (n_b) is not a parameter that affects thermodynamic, structural, and dynamical properties. In path-integral computer simulations, one chooses a “large” value of n_b so that all these properties converge, i.e., they do not change upon further increase in n_b . The value $n_b=10$ used in our work was chosen because we find no further changes in $E(T)$, $P(T)$, and $D(T)$ for $n_b>10$ (see, e.g., the response to point (8) below).

8) I would also like to request the authors to provide the details of how MSD varies with h , T/h , and n_b and whether taking into account only species A (or B) particles vs. all the particles make any difference in the conclusions.

In the left column of Fig. C below, we include the MSD(t) of the A-type atoms of the LJBM for different values of h and T . Panel (a) shows the MSD(t) of the A-type atoms as a function of time for the classical and quantum LJBM with $h=h_a, h_b, h_c$, and $T=0.6, 1.0$. These results are consistent with the results of Markland et al. [Markland, T. E. et al. Quantum fluctuations can promote or inhibit glass formation. Nature Physics 7, 134–137 (2011)]. Panel (b) shows the MSD(t) for $h/T=0.08$ from PIMD simulations performed at ($T=1.0, h=h_b$) and ($T=2.0, h=h_c$). The MSD's do not collapse onto a single master curve (see also response to point (5) above). In panel (c), we show the MSD(t) of the A-type atoms at ($T=0.8, h=h_c$) and for $n_b=10, 20, 40$. The MSD(t) for $n_b=10, 20, 40$ practically overlap with one another.

The right column of Fig. C shows the same results as in panels (a)(b)(c) for the case where all the atoms in the LJBM (A- and B-type atoms) are included in the calculation of the MSD. Using the A-type particles and all the particles in the calculation of the MSD(t) leads to similar conclusions.

Fig. C. (a) MSD of the A-type atoms in the classical and quantum LJBM with $h= h_a, h_b$, and h_c (orange, green, red, and purple, respectively). Temperatures are $T=0.6$ (solid lines) and 1.0 (dashed lines). (b) MSD of the A-type atoms for $h/T=0.08$. Orange

and blue lines correspond to $(T=1.0, h=h_b)$ and $(T=2.0, h=h_c)$. (c) MSD of the A-type atoms of the LJBM for $h=h_c$ and $T=0.8$, and for different number of beads per ring polymer, n_b . (d)(e)(f) Same as panels (a)(b)(c) for the case where all the atoms in the LJBM (A- and B-type particles) are included in the calculation of the MSD.

9) *It will be useful to see how the characteristic temperatures e.g. the onset temperature T_{onset} , the VFT divergence temperature T_0 and the Kauzmann temperature T_k depends on the degree of quantumness, h and n_b . This will enable direct comparison with experiments such as Ref. 14. It will also offer new insights to the ongoing debates about whether the ideal glass transition should occur at zero or finite temperatures [see e.g. Berthier et al., Nat. Commun. 10, 1508 (2019); Nath and Sengupta, J. Chem. Phys. 161, 034504 (2024)]. In Fig. 5, the authors have noted that T_k increases with increasing h . What could be the physical explanation for this trend?*

We thank the referee for bringing this point to our attention. We focus on the dependence of two characteristic temperatures, the Kauzmann temperature T_K and the VFT temperature T_{VFT} (as explained in point (5) above, the role of n_b does not affect our results).

The left panel of Fig. D shows the fit to the diffusion coefficient $D(T)$ using the VFT equation,

$$D(T) = D_0 \exp\left[-\frac{A}{T-T_{VFT}}\right] \quad (S10)$$

For all the values of h studied, the diffusion coefficient of the LJBM obeys Eq. S10. The corresponding values of $T_{VFT}(h)$ are included in the right panel of Fig. D together with the values of Kauzmann temperature $T_K(h)$ reported in the main manuscript. The values of $T_{VFT}(h)$ and $T_K(h)$ are remarkably close to one another.

Both $T_{VFT}(h)$ and $T_K(h)$ increase approximately linearly with increasing quantumness, for the values of h studied here. However, as shown in [Markland, T. E. et al. Quantum fluctuations can promote or inhibit glass formation. Nature Physics 7, 134–137 (2011)], further increase in h leads to a change in the dynamics of the LJBM where the diffusion coefficient *increases* with increasing h . Accordingly, both $T_{VFT}(h)$ and $T_K(h)$ are expected to *decrease* with increasing h , for $h>h_c=1.0000$. The role of quantum fluctuations on the dynamics of the LJBM is discussed in detail in Refs. [Markland, T. E. et al. Quantum fluctuations can promote or inhibit glass formation. Nature Physics 7, 134–137 (2011)., Markland, T. E. et al. Theory and simulations of quantum glass forming liquids. Journal of Chemical Physics 136, 074511–074511 (2012).]. In these works, the authors argue that the non-monotonic dependence of D with h (for a constant T) is due to a competition between the atoms' delocalization (induced by quantum fluctuations) and caging effects (due to the nearest neighbors of the atoms) which can enhance or suppress the diffusion of the LJBM atoms, depending on h .

Fig. D. (a) Diffusion coefficient as a function of temperature for the classical and quantum LJBM studied. Lines are fits using the VFT equation, Eq. S10. (b) VFT temperature $T_{VFT}(h)$ obtained from the fits in (a). For comparison, the Kauzmann temperature $T_K(h)$ is also included. The values at $h=0$ are for the classical LJBM (MD simulations). Both $T_{VFT}(h)$ and $T_K(h)$ exhibit the same qualitative behavior.

We added Fig. D and the discussion above in the last section of the new Supplementary Information.

10) *At a conceptual level, the physical meaning of tuning the Planck's constant h is not very clear to me. The spring constants in the second term of the Hamiltonian in Eqn. 4 is a function of T/h . Does it imply that T/h should be the control parameter instead of T and h independently? Specifically, is there any data collapse of E_{IS} , E_{vib} in Figs 2 or 4, Sc in Fig 5 and D in Fig 7 plotted against T/h ? And more generally, by some combination of T , h and n_b ?*

As noted in the response to point (7) above, n_b is not a control parameter since our results already converged with n_b (i.e. they do not change upon further increase in n_b).

Fig. B shows that, in general, the LJBM properties do not collapse onto a master curve that is solely a function of T/h . Indeed, most of the thermodynamic and dynamic properties of the LJBM should depend on both T , T/h , and/or h . This can be understood by the partition function given in Eq. 3 of the main manuscript; while the Hamiltonian [Eq. 4] depends only on T/h , the partition function $Q(N,V,T)$ depends on both $\beta = 1/k_B T$ and T/h . This implies that, for example, thermodynamic properties that involve partial derivatives of $Q(N,V,T)$ with respect to V and T will have different dependence on $(T/h, T)$.

11) *One of the key assumptions in the present analysis is that the distribution of IS minima with energy e_{IS} should be independent of h . The authors have argued that this should be the case based on the numerical observation of the collapse of ring-polymers in the IS, [Refs 41, 42]. However, this implies that the parameters $\{\alpha, E_0, \sigma^2\}$ should be independent of h as well. This is however, in contradiction to the evidence shown in earlier works, e.g. inset of Fig. 4a in Ref. 41. Perhaps the authors consider sharpening the explanations to rationalize the choice of option B in sec 3.3.*

The point raised by the reviewer is subtle and very important. Indeed, one of the main points of Sec. 3.3 is to address this issue. As explained there, in our first manuscript focused on the PEL of quantum liquids [41], we applied Eq. 15 (for Gaussian PELs) without any assumption on the parameters $\{\alpha, E_0, \sigma^2\}$. Hence, to fit the PIMD simulation data for the model liquid studied in Ref. [41], we had to allow these quantities

to vary with h . In the current manuscript, we argue that the PEL parameters $\{\alpha, E_0, \sigma^2\}$ should be independent of h . The rationale for this is that for every system we have studied (since Ref. 41 was published), over wide ranges of T and P , the ring-polymers always collapse at the IS, implying that $\{\alpha, E_0, \sigma^2\}$ should be h -independent. This approach is also strongly supported by Fig. 5 which shows that the configurational entropy obtained by assuming that $\{\alpha, E_0, \sigma^2\}$ are h -independent, is self-consistent within the PEL formalism.

In the new version of the manuscript, we clarify the point raised by the referee. The following discussion is added in page 12: “...(A) One may assume that the quantities $\{\alpha, E_0, \sigma^2\}$ vary with the quantumness of the liquid (as quantified by h), but the IS of the RP-PEL with non-collapsed ring-polymers are rare, difficult to sample in the RPMD simulations. This is the approach followed originally in Ref. [42]. Alternatively, one may consider that (B) there are no IS in the RP-PEL where ring-polymers are not collapsed at the working conditions and hence, the quantities $\{\alpha, E_0, \sigma^2\}$ are independent of whether the liquid obeys classical or quantum mechanics. In this work, we depart from the approach followed in Ref. [42] and assume that option (B) holds... “

12) Even for purely classical computations, the Planck's constant h do appear in the total entropy calculation through the ideal gas contribution. Thus for computational purpose h is non-zero even for classical systems e.g. $h \sim 0.18$ in Argon units. There is a possibility that the total entropy may become negative at finite temperatures for increasing h , violating the third law of thermodynamics. Hence, I would like to request for total entropy data for different h (or T/h). This may easily be computed by standard techniques such as thermodynamic integration, see e.g. Nandi et al. J. Chem. Phys. 156, 014503 (2022) for a critical assessment of the method.

We thank the referee for raising this important point. Following the reviewer's suggestion, we include in the SI of the manuscript, Fig S4(b), the total entropy of the LJBM, $S(T)$, for the different values of h studied. We also clarify the units used in this work. We find that $S(T) > 0$ for all values of T and h studied. The following new section is added to the SI:

III. Entropy of the Classical and Quantum LJBM

The total entropy for the studied LJBMs as a function of temperature $S(T)$ for $V = 9.4$ is obtained by thermodynamic integration as performed in Refs. [5,6] for the classical LJBM. Briefly, for each of the classical/quantum LJBMs considered, characterized by a given value of h , we follow the two-step process detailed below.

(i) In the first step, we evaluate the entropy of the target LJBM at the reference state $T_0 = 5.0$ and $V_0 = 9.4$ (in reduced units); V_0 is the volume studied in the main manuscript. To do so, we compress the LJBM from a very large volume $V_{ig} = 3 \times 10^5$ to the target volume V_0 along the isotherm, $T_0 = 5.0$. From the thermodynamic relationship $dE = T dS - P dV$, one obtains the following expression,

$$S(T_0, V_0) = S(T_0, V_{ig}) + \frac{1}{T_0} [E(T_0, V_0) - E(T_0, V_{ig})] - \frac{1}{T_0} \int_{V_0}^{V_{ig}} P(T_0, V) dV \quad (S2)$$

The volume V_{ig} is very large and the temperature T_0 is very high for all the LJBMs considered; at these conditions, $h/\sqrt{2\pi m_A k_B T_0} \ll V_{ig}^{1/3}$ for all the values of h studied. Accordingly, at (T_0, V_{ig}) , the (quantum and classical) LJBM can be approximated by an ideal gas binary mixture. Hence, $E(T_0, V_{ig})$ and $S(T_0, V_{ig})$ in Eq. S2 are given by the

energy and entropy of the ideal gas binary mixture, i.e., $E(T_0, V_{ig}) = E_{ig}(T_0, V_{ig}) = \frac{3}{2} N k_B T_0$ (where $N = N_A + N_B$) and $S(T_0, V_{ig}) = S_{ig}(T_0, V_{ig})$; the same expressions hold for both the quantum and classical ideal gas binary mixtures since $h/\sqrt{2\pi m_A k_B T_0} \ll V_{ig}^{1/3}$ [3,7]. The entropy of a (classical or quantum) ideal gas binary mixture at (T, V) is given by $(m_A = m_B)$ [6],

$$\frac{S_{ig}(T,V)}{Nk_B} = -\frac{N_A}{N} \ln\left(\frac{N_A}{N}\right) - \frac{N_B}{N} \ln\left(\frac{N_B}{N}\right) + \frac{3}{2} \ln\left(\frac{2\pi m_A V^{\frac{2}{3}}}{\beta h^2}\right) - \ln\left(\frac{N}{e^{\frac{5}{2}}}\right) \quad (\text{S3})$$

In this expression, all quantities are given in real units. Accordingly, to obtain $S_{ig}(T_0, V_0)$ we substitute $T = T_0 \times (\epsilon_{AA}/k_B)$ and $V = V_0 \times (\sigma_{AA}^3)$ in Eq. S2 (the reduced units of T and V are $[T] = \epsilon_{AA}/k_B$ and $[V] = \sigma_{AA}^3$). For the classical LJBM, $h = 6.62607015 \times 10^{-34} \text{ J s}$ in Eq. S2. For the quantum LJBM characterized by Planck's constants $h_a = 0.2474$, $h_b = 0.5000$ and $h_c = 1.0000$ in reduced units, $[h] = \sigma_{AA} (m_A \epsilon_{AA})^{1/2}$, the value of h in Eq. S2 is given by $h = h_x \times [\sigma_{AA} (m_A \epsilon_{AA})^{1/2}]$ (where $x = a, b, c$). Note that the right-hand-side of Eq. S2 depends only on h_x and is independent of the specific values of $(m_A, \sigma_{AA}, \epsilon_{AA})$ considered. The values of $S_{ig}(T_0, V_{ig})/Nk_B$ obtained from Eq. S2 are included in Table S1

System	$h \text{ [J s]}$	$S_{ig}(T_0, V_{ig})/Nk_B$	$E(T_0, V_0)/N$ [ϵ_{AA}]	$S(T_0, V_0)/Nk_B$
Classical	6.62607×10^{-34}	22.1628	5.8399	13.5865
$h_a = 0.2474$	2.59669×10^{-33}	18.0619	5.8759	9.5120
$h_b = 0.5000$	5.24796×10^{-33}	15.9512	6.0542	7.4084
$h_c = 1.0000$	1.04959×10^{-32}	13.8718	7.0975	5.4021

Table S1 Values of the Planck's constant studied in the main manuscript given in reduced units (first column). For comparison, included in the second column are the corresponding values of h in real units, obtained from the first column via the expression $h = h_x \times [\sigma_{AA} (m_A \epsilon_{AA})^{1/2}]$ and assuming $(m_A = 39.948 \text{ amu}, \sigma_{AA} = 1.0 \text{ \AA}, \epsilon_{AA} = 1.0 \text{ kJ/mol})$. $S_{ig}(T_0, V_{ig})$ is the entropy of the ideal gas binary mixture given by Eq. S3 at $T_0 = 5.0$ and $V_{ig} = 3 \times 10^5$ (both quantities given in reduced units). $E(T_0, V_0)$ and $S(T_0, V_0)$ are the energy and entropy of the LJBM studied at the reference state point $(T_0, V_0 = 9.4)$; $S(T_0, V_0)$ is evaluated using Eq. S2.

To evaluate Eq. S2, we perform additional MD/PI computer simulations at $T = T_0$ and different values of V in the range $[V_0, V_{ig}]$. Fig. S3 shows the excess pressure $P_{ex}(T_0, V) = P(T_0, V) - Nk_B T_0/V$ of the LJBM obtained from the MD/PI computer simulations. The values of $E(T_0, V_0)$ and $S(T_0, V_0)$ [Eq. S2] are given in Table S1.

Fig. S3. Excess pressure $P_{ex}(T_0, V)$ of the LJBM obtained from MD/PI computer simulations at $T_0 = 5.0$ and $V \geq V_0 = 9.4$. The dashed line is the first virial correction to the pressure calculated analytically, $B_2(T_0) k_B T_0 \left(\frac{N}{V}\right)^2$, with $B_2(T_0) = 0.53622$ (see Ref. [6]).

(ii) The second step to get $S(T, V_0)$ consists of a thermodynamic integration along the V_0 -isochore starting from the reference temperature T_0 . Specifically, since $dE = T dS$ along an isochore, one obtains the following expression,

$$S(T) = S(T_0) - \int_{T_0}^T \frac{C_v(T')}{T'} dT' \quad (\text{S4})$$

where $C_v(T) = \left(\frac{\partial E}{\partial T}\right)_V$ is the constant-volume heat capacity (for simplicity, from now on, we omit the dependence of all quantities on V_0). To evaluate $C_v(T)$, we perform MD/PI computer simulations at various temperatures ($V = V_0$) and calculate the total energy of the quantum LJBM, $E(T)$. For the classical case, we follow Ref. [5,6] and use Tarazona's approximation [4] for the potential energy, $U(T) = a_0 + b_0 T^{5/3}$ (where a_0 and b_0 are constants). Accordingly, for the classical LJBM, we interpolate $E(T)$ using the expression for the total energy $E(T) = \frac{3}{2} N k_B T + a_0 + b_0 T^{5/3}$. For the quantum LJBM, the values of $E(T)$ cannot be fitted using this approach. In these cases, $E(T)$ is fitted using a fifth-order polynomial $E(T) = \sum_{i=0}^5 a_i T^i$. Fig. S4(a) shows the $E(T)$ obtained from the MD/PI computer simulations together with the corresponding fifth-order polynomial fit. The entropy $S(T)$ is evaluated analytically using Eq. S4 and is included in Fig. S4(b).

Fig. S4. (a) Energy of the classical and quantum LJBM as a function of temperature ($V = 9.4$) obtained from MD and PI computer simulations with different values of Planck's constant h (symbols). Lines are the polynomial fittings to the symbols [see text]. (b) Entropy of the LJBM obtained from (a) based on thermodynamic integration, Eq. S4.

In Fig. S4 (and Fig. 2 (c)(d) of the manuscript), the temperature $T = 0.55$ for $h = hc$ (purple lines) was not included in the fitting because, at this temperature, the values of $D(T)$ [Fig. 1(c)] and $E_{IS}(T)$ [Fig. 3(a)] deviate from the corresponding trend observed at higher T ($h = hc$), probably due to poor equilibration. Nonetheless, even for ($h = hc$, $T = 0.55$) (purple line in Fig. S4), we find that the entropy is positive, $S = 0.225$. Summarizing, as shown in Fig. S4, $S(T) > 0$ for all the LJBM studied (as expected). We confirm that the results in are not sensitive to alternative (reasonable) fits to $E(T)$.

References

- [4] Y. Rosenfeld and P. Tarazona. *Density functional theory and the asymptotic high density expansion of the free energy of classical solids and fluids*. Molecular Physics 95, 141–150 (1998).
- [5] S. Sastry. *The relationship between fragility, configurational entropy and the potential energy landscape of glass-forming liquids*. Nature 409, 164–167 (2001).
- [6] F. Sciortino, W. Kob, and P. Tartaglia. *Thermodynamics of supercooled liquids in the inherent-structure formalism: A case study*. Journal of Physics: Condensed Matter, 12, 6525–6534 (2000).
- [7] M. E. Tuckerman. *Statistical Mechanics: Theory and Molecular Simulation*. Oxford Graduate Texts. Oxford University Press, Incorporated, Oxford, 2nd ed edition, 2023.

13) *Is the $S_C(T)$ in Fig. 7 averaged over IS energies sampled at a given temperature? This may be clarified.*

We thank the reviewer for bringing this point to our attention. To address the reviewer's question, we add the following text in the beginning of the second-to-last paragraph of Sec. 4: "Fig. 6 (Fig. 7 in the previous version of the manuscript) shows $S_{IS}(T)$ for the classical and quantum LJBM studied. For the *classical* LJBM, the values of $S_{IS}(T)$ in Fig. 6 are obtained from thermodynamic integration, following the procedure of Refs. [43,63] ($S_{IS}(T)$ is shown in Fig. S5(a) of the SI). For the *quantum* LJBM, the values of $S_{IS}(T)$ in Fig. 6 are taken from the corresponding expression in the PEL formalism, Eq.(10), and using the equilibrium IS energy [$e_{IS} \rightarrow E_{IS}(T)$] given by Eq. (28). The resulting expressions for $S_{IS}(T)$ are evaluated at the same temperatures at which the diffusion coefficient $D(T)$ is calculated from the MD/PI computer simulations. The diffusion..."

Motivated by the reviewer's question, in the new version of the Supplementary Information (SI) we also show that the "theoretical" values of $S_{IS}(T)$ for the quantum LJBM, obtained from Eqs. (10) and (28), are in good agreement with the values of $S_{IS}(T)$ calculated "numerically" via thermodynamic integration and PI computer simulations. Specifically, the following section is added to the SI:

"IV. Configurational Entropy

In this section, we discuss how the configurational entropy $S_{IS}(T)$ of the classical and quantum LJBM are obtained numerically, using MD/PI computer simulations and thermodynamic integration. As shown in Fig. 4(b) of the main manuscript, for the *quantum* LJBM, the values of $S_{IS}(T)$ evaluated numerically (symbols) are in very good agreement with the corresponding predictions from the PEL formalism [Eqs. (10) and (28)] (solid lines).

The numerical procedure to evaluate $S_{IS}(T)$ based on thermodynamic integration has been used in the past to calculate the configurational entropy of classical atomistic and molecular liquids, including the classical LJBM [5,6] and water [1,2]. This numerical approach is based on the concept of vibrational entropy, S_{vib} , defined implicitly via the expression

$$S(N, V, T) \equiv S_{IS}(N, V, T) + S_{vib}(N, V, T) \quad (S5)$$

To evaluate $S_{IS}(T)$, one calculates $S(T)$ and $S_{vib}(T)$ independently. The calculation of $S(T)$ for the classical and quantum LJBM studied is performed by thermodynamic integration, as explained in Sec. III above. To calculate $S_{vib}(T)$, one evaluates the corresponding expression from the PEL formalism using MD/PI simulations, as explained next.

In the PEL formalism (for a fixed composition, $N_A = 4 N_B$ and $N = N_A + N_B$), the Helmholtz free energy can be written as

$$F(N, V, T) = F_{IS}(N, V, T) + F_{vib}(N, V, T) \quad (S6)$$

where $F_{IS}(N, V, T) \equiv E_{IS}(N, V, T) - T S_{IS}(N, V, T)$ [see Eq. (13) of the main manuscript]. Since the Helmholtz free energy is $F(N, V, T) = E(N, V, T) - T S(N, V, T)$ and the vibrational energy is defined such that $E(N, V, T) \equiv E_{IS}(N, V, T) + E_{vib}(N, V, T)$, it follows from Eqs. S5 and S6 that

$$S_{vib}(N, V, T) = [E_{vib}(N, V, T) - F_{vib}(N, V, T)]/T \quad (S7)$$

(i) *Classical LJBM*. Following Refs. [5,6], for the classical LJBM, it can be shown that Eq. S7 leads to the following expression,

$$S_{vib}(N, V, T) = 3Nn_b k_B [1 - \ln(\beta \hbar \omega_0)] - k_B S(N, V, T, e_{IS}) \quad (S8)$$

Expression S8 assumes that the PEL of the classical LJBM is Gaussian and harmonic which is consistent with MD simulations [5,6] (see also Figs. 2(a) and 2(b) of the main manuscript). Our values of $S_{IS}(T)$ for the classical LJBM, obtained numerically using Eqs. S5 and S8, are shown in Fig. S5(a).

Following the procedure in Ref. [6], from the $S_{IS}(T)$ shown in Fig. S5(a), we obtain $S_{IS}(e_{IS})$. The symbols in Fig. S5(b) correspond to the resulting $S_{IS}(e_{IS})$ evaluated at selective values of e_{IS} . The line in Fig. S5(b) is the prediction from the PEL formalism based on the Gaussian approximation, Eq. (10). Following Ref. [5], in the classical case, we obtain E_0 and σ^2 from the fit in Fig. 2(a) [orange line] based on Eq. (15); the parameter α is fit to maximize the overlap of the numerical values of $S_{IS}(e_{IS})$ (symbols) and the solid line in Fig. S5(b) [Eq. (10)].

Fig. S5. (a) Configurational entropy $S_{IS}(T)$ of the classical LJBM as a function of temperature. $S_{IS}(T)$ is calculated numerically following the procedure of Refs. [5,6] via thermodynamic integration and MD simulations. (b) $S_{IS}(e_{IS})$ obtained from (a) for selected values of e_{IS} (symbols); see Ref. [6]. The line is the prediction from the PEL formalism based on the Gaussian approximation, Eq. 10. In Eq. 10 the parameters E_0 and σ^2 are obtained from Fig. 2(a) using Eq. 15; the parameter α is fit to maximize the overlap of the numerical values of $S_{IS}(e_{IS})$ (symbols) and the solid line [Eq. 10]. We obtain $E_0 = 7304.012$, $\sigma^2 = 134.223$, and $\alpha = 0.988$.

(ii) *Quantum LJBM*s. Using Eqs. 27 and 29 of the main manuscript in Eq. S7 leads to the following expression for the quantum LJBM

$$S_{vib}(N, V, T) = 3Nn_b k_B [1 - \ln(\beta \hbar \omega_0)] - k_B [\mathcal{S}(N, V, T, E_{IS}) + \tilde{B}_0(N, V, T) + \tilde{B}_1(N, V, T) E_{IS}(N, V, T)] + \frac{1}{T} \left[\left(\frac{\partial \mathcal{S}}{\partial \beta} \right)_{N, V, E_{IS}} + \left(\frac{\partial \tilde{B}_0}{\partial \beta} \right)_{N, V} + \left(\frac{\partial \tilde{B}_1}{\partial \beta} \right)_{N, V} E_{IS}(N, V, T) \right] \quad (\text{S9})$$

Most of the quantities needed in Eq. S9 (specifically, the PEL variables $\{\alpha, E_0, \sigma^2, a, b, c_{0,1}, c_{0,2}, c_{1,0}, c_{1,1}, c_{1,2}\}$) are evaluated in the main manuscript. The only missing quantity in Eq. S9 is $c_{0,0}$. Therefore, using Eqs. S5 and S9, one can evaluate $S_{IS}(T)$ up to the unknown (additive) T-independent quantity $c_{0,0}$. The symbols in Fig. 4(b) of the main manuscript are the values of $S_{IS}(T)$ for the quantum LJBM

obtained numerically following the procedure described here. For each value of h , the additive constant $c_{0,0}$ (for fix $V = 9.4$) is fit to maximize the overlap with the PEL predictions (lines in Fig. 4(b)). The agreement with the theoretical predictions based on the PEL formalism is very good. For comparison, Fig. S6(a)-(c) shows the same values of $S_{IS}(T)$ included in Fig. 4(b) of the main manuscript but expressed as a function e_{IS} , by using Eq. (28) (symbols). The agreement in Fig. S6(a)-(c) between the numerical values of $S_{IS}(e_{IS})$ (symbols) and the PEL predictions based on the Gaussian approximation, Eq. (10) (lines), are very good for all the classical/quantum LJBM studied.

Fig. S6. (a) Configurational entropy as a function of the IS energy, $S_{IS}(e_{IS})$, for the quantum LJBM with Planck's constant (a) $h = h_a$, (b) $h = h_b$, and (c) $h = h_c$. Symbols are the values of $S_{IS}(e_{IS})$ obtained numerically, based on Eqs. S5 and S9, and thermodynamic integration. The line is the expression for $S_{IS}(e_{IS})$ based on the Gaussian approximation of the PEL, Eq. (10), taken from Fig. S5(b).

The vibrational contributions to the total entropy for the LJBM studied, $S_{vib}(T)$, are shown in Fig. S7. For all the LJBM studied, $S_{vib}(T)$ decreases monotonically upon cooling. Curiously, while $S_{vib}(T) > 0$ for the classical LJBM and quantum LJBM with $h = h_a$ and h_b , for the most quantum LJBM ($h = h_c$), we find that $S_{vib}(T)$ becomes slightly negative at the lowest temperatures accessible in our PI simulations, at approximately $T < 0.65$. We note, however, that $S_{vib}(T)$ is a quantity defined within the PEL formalism, with no direct physical interpretation. Indeed, $S_{vib}(T) \equiv S(T) - S_{IS}(T)$ and hence it is the difference between two positive quantities. In the limit $T \rightarrow 0$, $S(T) \rightarrow 0$ but $S_{IS}(T)$ may remain positive. This can be shown by considering a single quantum atom (ring-polymer) in a symmetric double-well potential with a finite energy barrier. In this case, $S(T) \rightarrow 0$ as $T \rightarrow 0$ but $S_{IS}(T) = k_B \log 2$, resulting in $S_{vib}(T) < 0$ at low temperatures.”

Fig. S7. PEL vibrational entropy $S_{vib}(T) \equiv S(T) - S_{IS}(T)$ associated to the LJBM studied. $S_{vib}(T)$ decreases monotonically upon cooling and remains positive except for the most quantum LJBM studied ($h = h_c$; purple line) at very low temperatures (approximately $T < 0.65$)."

- The new Fig.4 in the main manuscript, with the modified caption, are reproduced below.

Fig. 4 (a) Configurational entropy as a function of the IS energy, $S_{IS}(e_{IS})$, for the classical and quantum LJBM studied [Eq. 10]; $S_{IS}(e_{IS})$ is considered to be independent of the quantum character of the LJBM considered (see text). $S_{IS}(e_{IS})$ is obtained by thermodynamic integration and classical MD simulations of the LJBM, following the procedure of Refs. [43,64] (see Sec. III and IV of the SI). (b) Configurational entropy as a function of temperature, $S_{IS}(T)$. The dashed line is the $S_{IS}(T)$ for the *classical* LJBM (see Sec. III and IV of the SI). The $S_{IS}(T)$ for the quantum LJBM (solid lines) are obtained from (a) [Eq. 10] and substituting $e_{IS} \rightarrow E_{IS}(T)$ using Eq. (28). The symbols are the values of $S_{IS}(T)$ obtained numerically from PI computer simulations and thermodynamic integration, as explained in Sec. III and IV of the SI. The numerical results for $S_{IS}(T)$ (symbols) are in very good agreement with the corresponding values predicted by the PEL formalism (solid lines).

The following paragraph is added above Eq. (34): "To validate our results in Fig. 4(b), we also evaluate $S_{IS}(T)$ numerically, by combining PI computer simulations and thermodynamic integration. This procedure to evaluate $S_{IS}(T)$ is based on the definition of $S_{IS}(T)$ within the PEL formalism, $S_{IS} = S - S_{vib}$, where S_{vib} is the

vibrational entropy of the system; this procedure has been implemented in the past for calculate the $S_{IS}(T)$ of classical liquids, including atomistic model liquids and water [39,40,43,64]. The details of these calculations are included in the Sec. III and IV of the SI. The numerical values of $S_{IS}(T)$ for the quantum LJBM studied are indicated by the symbols in Fig. 4(b), and are in very good agreement with the predictions of the PEL (lines).

- We also moved Fig. 3 of the main manuscript to the SI.

14) Using $\ln Q$ instead of $c(T)$ in Eq. 34 and using different symbols to represent different h in Figs. 1, 2, 4, 7 may improve readability.

We follow the reviewer's suggestion and change the symbols in Figs 1, 2, 4, 7, as suggested. Regarding Eq. 34, we prefer to leave " $c(T)$ " to emphasize that $\ln(Q)$ can be treated as an auxiliary constant in the analysis.

15) For the sake of completeness, the formula to compute pressure P in Fig. 1 should be defined.

The estimator for the pressure used in our path-integral computer simulations is given by Eq. 12.8.43 of Ref. [Tuckerman, M. E. Statistical Mechanics: Theory and Molecular Simulation 2nd edn. Oxford Graduate Texts (Oxford University Press, Incorporated, Oxford)]. We include this information in the manuscript (paragraph 4 of Sec. 2, line 8: "**The pressure is evaluated using the estimator given in Eq. (12.8.43) of Ref. [56]**")

Reviewer #2 (Remarks to the Author):

The manuscript presents a rigorous extension of the potential energy landscape (PEL) formalism developed for classical liquids to quantum liquids, where at the quantum level, each particle is represented by a ring polymer. The system studied is the well known Lennard-Jones binary mixture, the Kob Andersen model. This work is built upon their prior foundational work, Refs 13,41,42, where they studied the PEL of quantum liquids. In the present study, they have extended the formalism to study the configurational entropy and investigate the validity of the Adam-Gibbs relationship between diffusion and configurational entropy. The important observations are the following. The configuration entropy expression of the classical and quantum liquids remains the same because, in the inherent state, the ring polymer collapses to a point (shown in the earlier studies for different systems). However, for the quantum system, the inherent structure energy at a particular temperature is dependent on Planck's constant 'h'. They also show that for high values of 'h', the anharmonic contributions of the PEL become important. Since the inherent structure energy is different for different values of 'h', the temperature dependence of the configurational entropy is different for different quantum systems. As the quantumness 'h' is increased, the configurational entropy disappears at a higher temperature. They have also shown that for all the systems, the Adam-Gibbs relationship holds spanning data over multiple orders of magnitude. This is an important observation as it clearly shows that the dynamics of a quantum liquid depends on the available inherent energy states at that temperature.

The use of ring polymer molecular dynamics (RPMD) combined with PEL theory for quantum liquids is still relatively unexplored, giving the work a high degree of originality. The variation of the configurational entropy and the Adam-Gibbs relationship with h are interesting results and are shown for the first time. This study opens a new direction where the theories of glassy transition can now be applied to quantum systems.

Thus, I would like to state that this work is highly original and contributes a new direction for applying glass transition theories to quantum systems, a domain that has been largely underexplored. The methodology is rigorous, and the simulation parameters and analytical assumptions are clearly stated. The theoretical framework is well constructed, and the numerical consistency checks (e.g., Eq. 33 validation) add credibility to the conclusions.

I fully support the acceptance of this paper in Nature Communications. However, I would like the Authors to have some discussion on the following points.

We appreciate the reviewer for taking the time to read the manuscript thoroughly and for the encouraging feedback. We are pleased that the reviewer considers our work to be “highly original” and recognizes its potential to study the behavior of liquids close to their glass transition and in the presence of quantum effects (an area that, as the reviewer indicates, has been significantly underexplored).

1) Why does the configurational entropy decrease with h ? Numerically, we know that the effect comes from the inherent structure energy, but a few lines of discussion will be nice to have. Also, the Authors mention that “Our RPMD simulations show that S_{IS} shifts towards higher temperatures as h increases.” I think it will be better to state that “Our RPMD simulations show that at a particular temperature, the S_{IS} decreases with increasing h ”.

The reviewer raises an important and subtle point. (i) The decrease of $S_{IS}(T)$ with increasing h (a measure of the system quantumness) at a given temperature, is unexpected [Fig. 4(b)]. However, it is consistent with Ref. [Markland, T. E. et al. Quantum fluctuations can promote or inhibit glass formation. Nature Physics 7, 134–137 (2011)] where (ii) the diffusivity of the LJBM atoms surprisingly decreases as the quantumness of the system (h) increases [for values of h comparable to those studied in our work]. We are currently studying whether findings (i) and (ii) are general, valid to other monatomic systems; these results will be presented in a separate work.

In page 14, line 9 from the page bottom, we modify the text to read: “Our RPMD simulations show that $S_{IS}(T)$ shifts towards higher temperatures as h increases. In particular, the Kauzmann temperature T_K , defined as the temperature at which $S_{IS}(T) = 0$, increases considerably as the liquid becomes more quantum. Specifically, $T_K = 0.28, 0.32, 0.35, 0.40$ for classical, h_a, h_b, h_c respectively. This result is unexpected, since it suggests that the glass transition temperature increases as the system becomes more quantum. While surprising, our results are fully consistent with the work of Markland *et al.* [24,54] who find that the diffusivity of atoms in the LJBM studied here is suppressed as the system becomes more quantum [for low and intermediate values of h , comparable to the values studied in this work]. Overall, ...”

Regarding the sentence “Our RPMD simulations show that $S_{IS}(T)$ shifts towards higher temperatures as h increases”, we find that it is equivalent to the sentence proposed by the reviewer. Accordingly, we prefer leaving this sentence as it is since it flows more naturally with the discussion in the corresponding paragraph.

2) I would also like the Authors to discuss why the activation energy “A” in the Adam-Gibbs relationship increases with h and what it physically signifies.

Please see response to point (5) raised by reviewer #1.

Reviewer #1 (Remarks to the Author):

Review of the revised manuscript "Configurational Entropy and Adam-Gibbs Relation for Quantum Liquids" [NCOMMS-25-32290A] by Yang Zhou, Ali Eltareb, Gustavo E. Lopez, Nicolas Giovambattista

I have gone through the revised manuscript and the referee replies by Zhou et al. The revised version has a clearer exposition, additional details about the analysis and the results are now more convincing. Hence, I recommend its acceptance in Nature Communications. A few minor points are mentioned below:

1) In Fig. 1d, the legend for "classical" ($h=0$) should be put above h_a , similar to e.g. Fig 3a.

We thank the reviewer for bringing this point to our attention. The legend in Fig. 1d has been updated so that "classical" ($h=0$) appears above h_a , consistent with the legend ordering in the other figures (e.g., Fig. 3a).

2) Typo in line 690: $S_{IS}(T)$ in are  $S_{IS}(T)$ are.

We thank the reviewer for noticing this typo. It has been corrected in the revised version of the manuscript.

3) Last but not the least, several groups using diverse approaches have analysed the effect of anharmonic corrections to the potential energy landscape (PEL) and the configurational entropy, especially in the last few years and for the LJBM model. They have come to the same conclusion that the anharmonic correction to the configurational entropy is crucial to correctly describe glassy dynamics of classical liquids, in particular of the LJBM model. While it is reassuring that the authors find that it is also true for quantum liquids, the glass physics community sometimes are not aware of the importance of including the anharmonic corrections. Thus, these literatures should be acknowledged.

We thank the reviewer for raising this point. As suggested by the reviewer, we add the following sentence in the new version of the manuscript (Sec. 5, Summary and Discussion):

"We note that, in the case of classical liquids, recent studies found that anharmonic corrections to the configurational entropy are crucial to correctly describe the glassy dynamics of the LJBM [Das and Sastry, J. Non-Cryst. Solids: X 14, 100098 (2022); Ozawa et al., J. Chem. Phys. 151, 084504 (2019)]. Our results for the quantum LJBM with $h=h_b, h_c$ are consistent with these studies."

Reviewer #2 (Remarks to the Author):

As I mentioned earlier, the use of ring polymer molecular dynamics (RPMD) combined with PEL theory for Quantum liquids remain relatively unexplored, lending the work a high degree of originality.

The authors have answered all the questions raised by the first and second reviewers. However, in the manuscript and also in their replies to the questions, I feel that although what happens when quantumness is varied is well-documented, the physical understanding of the variation in quantumness remains unclear. Perhaps the authors could include a discussion in this direction.

We thank the reviewer for the positive assessment of our work and for raising this point. As the reviewer indicates, our study (i) documents quantitatively how the PEL properties

of the LJBM's vary with increasing quantumness (as quantified by the Planck's constant h), and (ii) the corresponding implications to the LJBM's thermodynamic and dynamical properties. However, our work does not explore the physical mechanisms underlying these variations, e.g., how the atoms in the system slow down at a given temperature when h increases. Understanding the underlying physical mechanisms that lead to the observed changes in the PEL and thermodynamic/dynamic properties of the LJBM's is important and deserves further investigation. Such a (non-trivial) connection will be addressed in a future study that we plan to submit for publication soon.

In the new version of the manuscript, we notice the point raised by the reviewer by adding the following sentence in the Summary and Discussion section (Sec. 5):

"We note that the PEL formalism for quantum liquids allows one to characterize quantitatively the changes in the thermodynamic and dynamical properties of a liquid as the quantumness of the system (as quantified by the Planck's constant h) is varied. However, the PEL formalism does not provide information on the physical mechanisms underlying such variation. For example, it remains unclear what physical mechanisms in the LJBM's studied result in the suppression of the atoms diffusivity and the increase of T_K when h increases. Addressing these issues is important and warrants further investigation."